# Evidence for a HURP/EB free mixed-nucleotide zone in kinetochore-microtubules

Cédric Castrogiovanni [1,2,7], Alessio V. Inchingolo [3,4,7], Jonathan U. Harrison [3,5], Damian Dudka [1,2,6], Onur Sen [3,4], Nigel J. Burroughs [3,5], Andrew D. McAinsh [3,4] ✉ & Patrick Meraldi [1,2] ✉

Current models infer that the microtubule-based mitotic spindle is built from GDP-tubulin with small GTP caps at microtubule plus-ends, including those that attach to kinetochores, forming the kinetochore-fibres. Here we reveal that kinetochore-fibres additionally contain a dynamic mixed-nucleotide zone that reaches several microns in length. This zone becomes visible in cells expressing fluorescently labelled end-binding proteins, a known marker for GTP-tubulin, and endogenously-labelled HURP - a protein which we show to preferentially bind the GDP microtubule lattice in vitro and in vivo. We find that in mitotic cells HURP accumulates on the kinetochore-proximal region of depolymerising kinetochore-fibres, whilst avoiding recruitment to nascent polymerising K-fibres, giving rise to a growing "HURP-gap". The absence of end-binding proteins in the HURP-gaps leads us to postulate that they reflect a mixed-nucleotide zone. We generate a minimal quantitative model based on the preferential binding of HURP to GDP-tubulin to show that such a mixed-nucleotide zone is sufficient to recapitulate the observed in vivo dynamics of HURP-gaps.

The microtubule-based mitotic spindle ensures faithful chromosome segregation during cell division. A key microtubule population are the kinetochore fibres (K-fibres) which are parallel arrays of -15 microtubules with their plus-ends engaged with kinetochores−multiprotein complexes that assemble on each centromere[1,2]. While the high dynamicity of free spindle microtubules allows them to rapidly explore 3D space, the sustained attachment of K-fibres to kinetochores enables microtubule polymerisation and depolymerisation forces to move chromosomes[3]. As a first approximation, K-fibre microtubules consist of GDP-tubulin lattices with a GTP-cap at the tip of the polymerising microtubule plus-ends[4,5]. Nevertheless, recent studies indicate that GTP-tubulin can also be punctually incorporated into the GDP-tubulin lattice, as part of repair mechanisms[6,7]. The GTP-cap is important for the stability of the microtubule and the probability of switching between polymerisation and depolymerisation (catastrophe). The potential for catastrophe is dependent on the equilibrium between GTP-cap growth, which occurs via the addition of new GTP-tubulin dimers, and GTP-cap loss which is a function of the rate of nucleotide hydrolysis. Loss of the GTP-cap exposes unstable GDP-tubulin lattice which triggers depolymerisation[8,9]. The mechanism of rescue (switch from depolymerisation to polymerisation) is less clear and may involve stochastic events at the tip or the presence of rescue sites within the lattice that favour the re-establishment of a GTP-cap[10]. This "dynamic instability" is apparent in the metaphase spindle, where bi-oriented chromosomes undergo quasi-periodic oscillations along the spindle axis as kinetochore-bound K-fibres alternate between periods of growth and shrinkage (Fig. 1A)[11].

[1]Department of Cell Physiology and Metabolism, Faculty of Medicine, University of Geneva, 1211 Geneva 4, Switzerland. [2]Translational Research Centre in Onco-hematology, Faculty of Medicine, University of Geneva, 1211 Geneva 4, Switzerland. [3]Centre for Mechanochemical Cell Biology, University of Warwick, Coventry, UK. [4]Division of Biomedical Sciences, Warwick Medical School, University of Warwick, Coventry, UK. [5]Mathematics Institute, University of Warwick, Coventry, UK. [6]Present address: Department of Biology, University of Pennsylvania, Philadelphia, PA, USA. [7]These authors contributed equally: Cédric Castrogiovanni, Alessio V. Inchingolo. ✉e-mail: a.d.mcainsh@warwick.ac.uk; patrick.meraldi@unige.ch

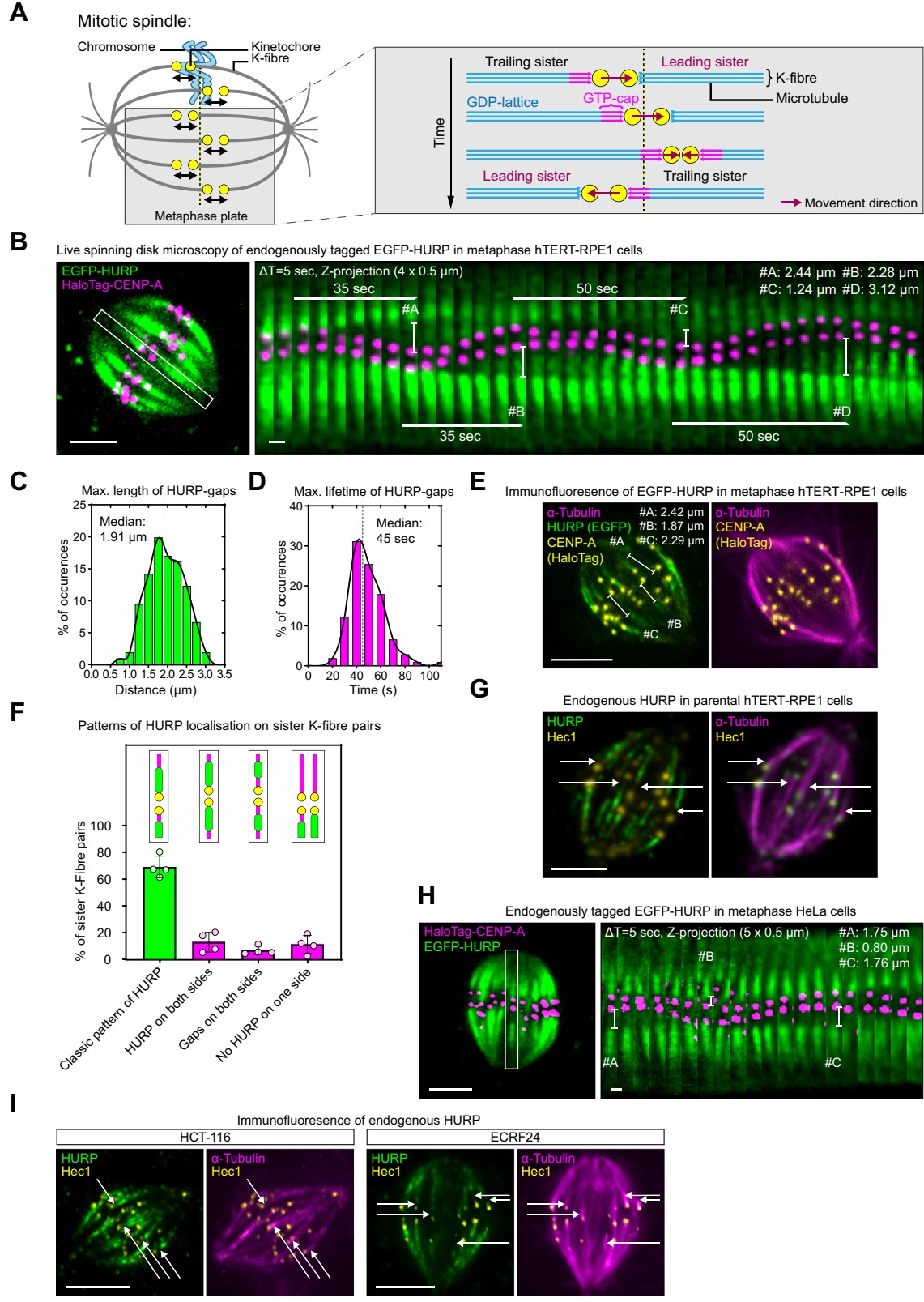

**A** Mitotic spindle:

**B** Live spinning disk microscopy of endogenously tagged EGFP-HURP in metaphase hTERT-RPE1 cells

**C** Max. length of HURP-gaps

**D** Max. lifetime of HURP-gaps

**E** Immunofluoresence of EGFP-HURP in metaphase hTERT-RPE1 cells

**F** Patterns of HURP localisation on sister K-fibre pairs

**G** Endogenous HURP in parental hTERT-RPE1 cells

**H** Endogenously tagged EGFP-HURP in metaphase HeLa cells

**I** Immunofluoresence of endogenous HURP

Based on structural data the GDP-tubulin lattice and the GTP-tubulin cap are thought to have different conformational states that are coupled to the nucleotide hydrolysis cycle[5,12,13]. Due to their preferential binding to GTP-tubulin the End-Binding (EB) proteins EB1 and EB3 bind the GTP-cap, while imposing a first structural change on the lattice towards the GDP-tubulin conformation[13,14]. In vitro experiments indicate that EB proteins further maintain their binding when the GTP nucleotide is hydrolysed but the γ-phosphate group has not yet been released[15], and that they will only dissociate once the microtubule lattice is in the GDP-tubulin form[13]. Therefore, EBs are used in vitro and in vivo as markers of the GTP-tubulin cap[13,14,16–19]. On K-fibres EBs accumulate on a narrow region of up to 100 nm, close to the kinetochore, consistent with the presence of a small GTP-cap[20,21].

**Fig. 1 | HURP is excluded from the K-fibre growth zone. A** Current model for sister-kinetochore oscillations in metaphase. **B** Live-cell imaging of metaphase hTERT-RPE1 EGFP-HURP/HaloTag-CENP-A cell. Inset shows sister K-fibre pair used for kymograph (right panel), bars display the HURP-gaps maximum length. **C, D** Distribution of live HURP-gap maximum lengths (**C**) and duration (**D** N = 4 independent experiments, n = 106 K-fibres in 29 cells) Black line = curve fit. **E** Immunofluorescence image of metaphase hTERT-RPE1 EGFP-HURP/HaloTag-CENP-A cell. Z-projection of 1 μm thickness. Bars show gap distances. **F** Quantification of mean HURP localisation patterns along sister K-fibre pairs in fixed cells. Data are presented as mean values ± SD (N = 4, n = 419 sister K-fibres in 19 cells). **G** Immunofluorescence image of endogenous HURP in metaphase hTERT-RPE1 cell. Z-projection of 0.5-μm thickness. Arrows indicate HURP-gaps. **H** Live-cell image of endogenously tagged EGFP-HURP in metaphase HeLa cell overexpressing HaloTag-CENP-A. Inset shows sister K-fibre pair used for kymograph (right panel), bars display gap maximum distances. **I** Immunofluorescence images of endogenous HURP in metaphase HCT-116 and ECRF24 cells. Z-projection of 1.5- and 1.0-μm thickness, respectively. Arrows indicate HURP-gaps. Scale bars = 5 μm and 1 μm (kymograph). Source data for all graphs are provided as a Source Data file.

The mitotic spindle contains tens of different microtubules-associated proteins (MAPs) that are distributed across the structure[4,22]. However, a notable exception is the hepatoma upregulated protein (HURP), which binds to K-fibres on a region of the K-fibres proximal to kinetochore—producing what is termed "HURP stripes"[23–25]. HURP stripes are the result of a spatial RanGTP gradient which originates at the chromosomes[26] and displaces Importin-β from one microtubule-binding site in HURP[23,25].

Here, we show that when analysing the spatiotemporal dynamics of endogenously tagged EGFP-HURP in human hTERT-RPE1 (non-transformed immortalised human retina pigment epithelial) cells expressing the kinetochore marker Halo-CENP-A, we observe a striking localisation in metaphase that does not fit the expected RanGTP directed pattern: as sister-kinetochore pairs oscillate along the spindle axis, a wide gap appears on the growing K-fibres between the HURP stripes and the attached kinetochore. Using both live-cell imaging, in vitro reconstitution and computational modelling approaches, we find that these "HURP-gaps" reflect the preferential binding of HURP to GDP-tubulin lattice and its relative exclusion from microtubules in the GTP-tubulin form. Our observation that the HURP-gap is neither HURP-nor EB-positive lead us to postulate that the HURP-gap reflects a region with a mixed-nucleotide content on K-fibre microtubules.

## Results

### HURP is excluded from growing K-fibre tips

We used endogenously tagged EGFP-HURP in human hTERT-RPE1 cells in conjunction with Halo-tagged CENP-A to study HURP dynamics during the quasi-periodic chromosome oscillations in metaphase. Kymographs of single oscillating sister-kinetochore pairs showed that when K-fibres switched from a depolymerising to the polymerising regime, HURP stripes on the growing K-fibre were seemingly left behind by the attached kinetochore, forming a region devoid of HURP which we term "HURP-gaps"; contrarily, no such gap was seen on the depolymerising K-fibres, with EGFP-HURP conterminous with the kinetochore (Fig. 1B and Supplementary Movies 1 and 2). The median maximal HURP-gap size on polymerising K-fibres measured 1.91 μm with a lifetime of ~45 s (n = 106; Fig. 1C, D). Immunofluorescence staining with anti-GFP and α-tubulin antibodies confirmed that HURP-gaps were present on 49.6 % of metaphase K-fibres (median width of 0.94 μm; Fig. 1E and Supplementary Fig. 1A, B), consistent with the notion that sister-kinetochores pairs are bound by one growing and one shrinking K-fibre. While most kinetochore pairs (69.2 ± 8%) displayed a HURP-gap on one side and a strong EGFP-HURP signal on the other side, 13 ± 7.4% had HURP signals on both sides, and 6.8 ± 3.3% had HURP-gaps on both sides (Fig. 1F and Supplementary Fig. 1C). These data are consistent with the fact that both sister kinetochores can be transiently connected to shrinking or growing K-fibres[27]. The HURP-gap was not a by-product of the GFP-tag or cell line since it was also visible after staining with antibodies directed against endogenous HURP (Fig. 1G) and visible in live images of HeLa (hypertriploid cervical carcinoma) cells expressing endogenously tagged EGFP-HURP (Fig. 1H and Supplementary Movie 3), as well as fixed HCT-116 (pseudo-diploid colorectal carcinoma) and ECRF24 (immortalised human umbilical vein endothelial) cells stained with HURP antibodies (Fig. 1I). These

data suggest that in the metaphase spindle, HURP is absent from nascent polymerising K-fibre regions connected to kinetochores, despite high RanGTP concentrations.

To quantify the spatiotemporal dynamics of HURP-gaps and kinetochore motion we used lattice light-sheet microscopy to acquire full 3D volumes of hTERT-RPE1 EGFP-HURP/Halo-CENP-A cells in two fluorescence channels every 4.14 s (Supplementary Movie 4 and Fig. 2A, B). 3D kinetochore tracking[28], and measurement of EGFP-HURP intensity along the associated K-fibre showed that as metaphase sister kinetochores oscillated back-and-forth, HURP was exclusively present on depolymerising K-fibres (associated to the leading kinetochore sister), and never on the growing K-fibre (associated to the trailing kinetochore sister) (Fig. 2C, D). As soon as a directional switch occurred, HURP started accumulating on the depolymerising K-fibre that was previously devoid of HURP, and a gap started to form on the opposite, growing K-fibre (Fig. 2C, D). The maximal HURP-gap size measured 1.56 μm in the average HURP profile from lattice light-sheet data consistent with the 1.91 μm maximum gap measured via spinning disc microscopy. Higher HURP intensities on depolymerising K-fibres correlated with a higher probability for a directional switch (Fig. 2E), hinting that HURP may act as a microtubule-rescue factor. We conclude that HURP-gaps are associated with newly assembled K-fibres.

### HURP-gaps on K-fibres may reflect the tubulin-nucleotide status

One reason for the formation of HURP-gaps on newly formed K-fibres sections could be a slow microtubule binding from the cytoplasmic HURP pool or slow diffusion from neighbouring HURP stripes. Previous fluorescence recovery after photobleaching (FRAP) experiments on the entire spindle showed that HURP is exchanged within 10–20 s with no substantial immobile fraction[24,25]. When we performed the same experiments on single K-fibres in hTERT-RPE1 EGFP-HURP/Halo-CENP-A cells, we again observed a fast exchange (~10 s—mobile fraction 91.7%) for EGFP-HURP, indicating that HURP is rapidly recruited to K-fibres (Fig. 3A, B). Since this value is fourfold lower than the median HURP-gap lifetime (~45 s, Fig. 1D) we excluded slow HURP recruitment as the cause of the HURP-gaps.

A second possibility is that HURP is unable to bind nascent K-fibre due to the tyrosination/detyrosination cycle. Indeed, detyrosinated tubulin is enriched in stable microtubules, such as K-fibre lattices, while newly assembled microtubules consist of tyrosinated tubulin[29–31]. We therefore treated metaphase-arrested hTERT-RPE1 EGFP-HURP/Halo-CENP-A cells for 45 min with DMSO or an inhibitor of the detyrosination (Parthenolide[32]), and monitored the HURP-gaps by live-cell imaging. If tubulin tyrosination prevented HURP binding, we would expect wider HURP-gaps. This was not, however, the case, even though Parthenolide treatment strongly decreased the level of detyrosinated tubulin (Fig. 3C, D and Supplementary Movie 5). We conclude that HURP is not excluded from growing K-fibres due to the tubulin detyrosination/tyrosination cycle.

A final possibility was that HURP is excluded from newly assembled microtubules due to the presence of GTP-tubulin, as compared to the "old" K-fibre lattice that consists of GDP-tubulin. To evaluate this possibility, we treated cells with increasing doses of the microtubule-stabilising agent taxol. While at low concentrations, taxol is thought to

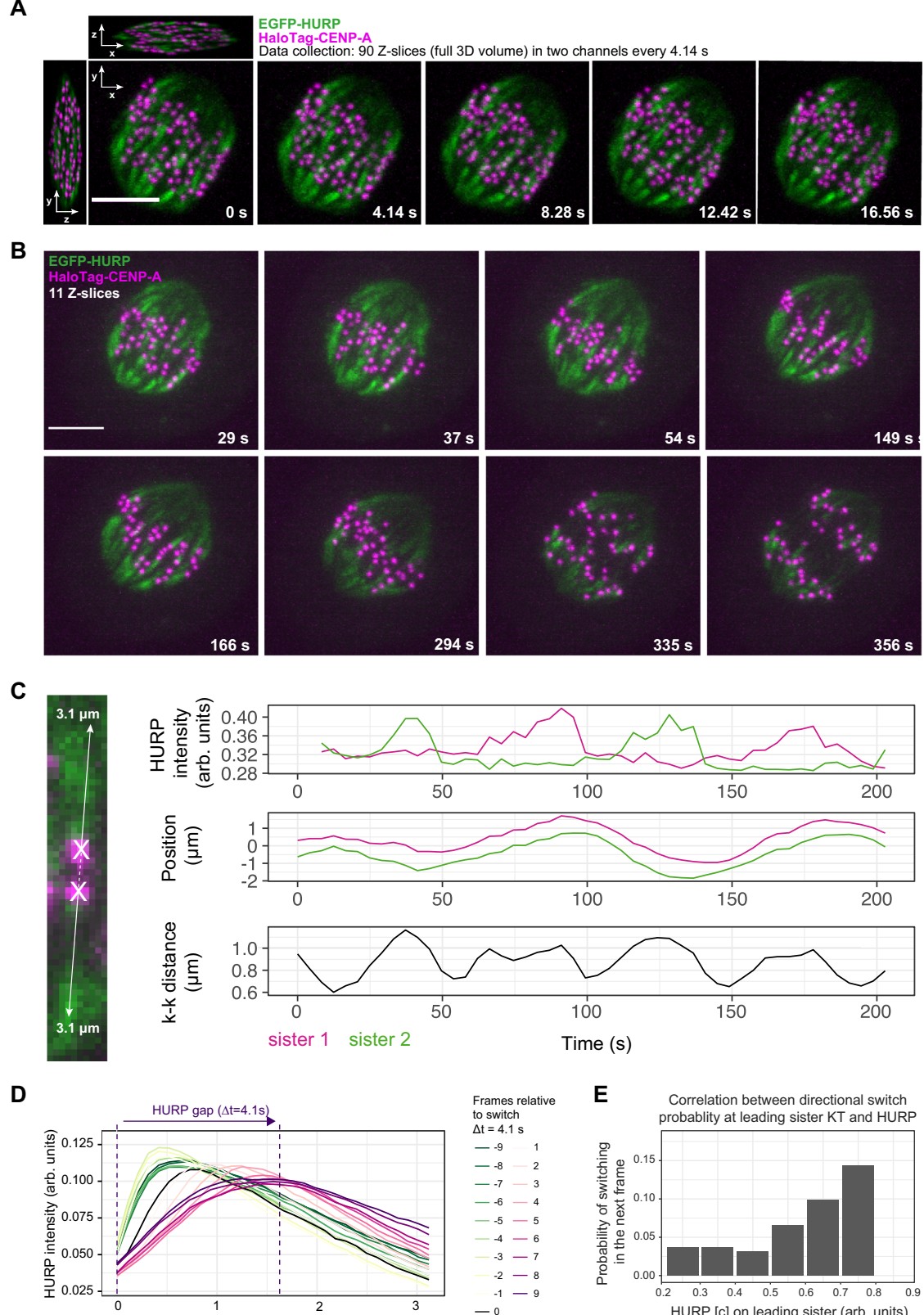

**Fig. 2 | HURP-gaps are precisely linked to K-fibre growth state. A** Lattice light-sheet 3D representation (X-, Y- and Z-projections) of a metaphase hTERT-RPE1-EGFP-HURP/HaloTag-CENP-A cell and relative movie stills projected in Z (90 z-slices). Scale bar = 5 μm. **B** Lattice light-sheet movie stills of the same cell in **A** projected in Z (11 z-slices). Scale bar = 5 μm. **C** Exemplary HURP intensities and sister-kinetochore positions in single pair over time, obtained after 3D kinetochore tracking. **D** Average spatial distribution of HURP along K-fibre over time relative to switch from leading to trailing (N = 3, n = 69 pairs). **E** Probability of directional switch in the next frame versus HURP on the K-fibre in the current frame (N = 3, n = 69 pairs). Source data for all graphs are provided as a Source Data file.

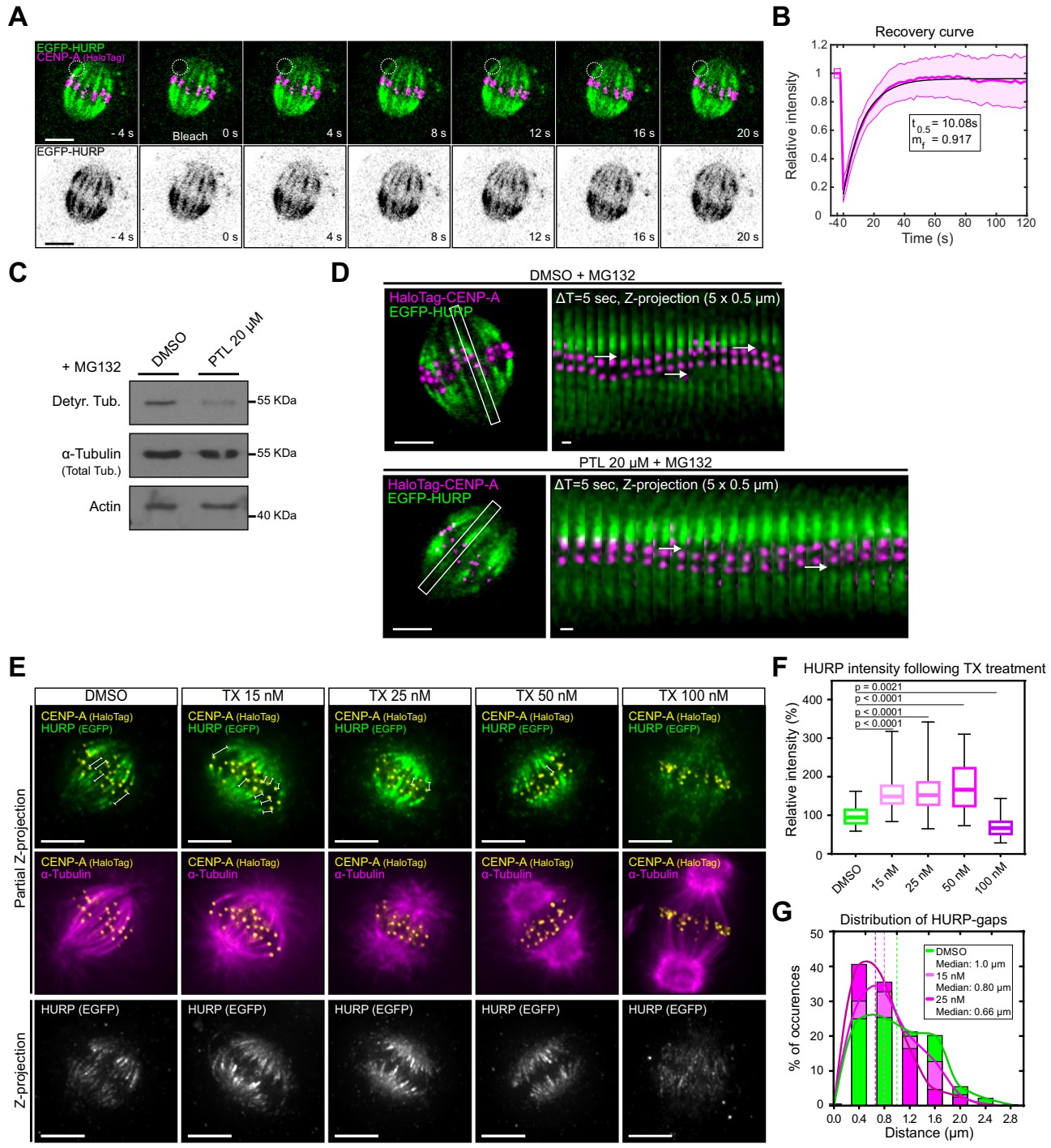

**Fig. 3 | HURP-gaps are precisely linked to K-fibre conformation. A** FRAP time-lapse of hTERT-RPE1 EGFP-HURP/HaloTag-CENP-A cell. Inset indicates bleached region on single K-fibre. **B** EGFP-HURP FRAP recovery curve. Data are presented as mean values ± SD. Black line = fit of mono-exponential recovery ($N = 4$, $n = 47$ K-fibres). **C** Western blot analysis of detyrosinated tubulin levels in hTERT-RPE1-EGFP-HURP/HaloTag-CENP-A protein extracts of cells blocked in metaphase (10 μM MG132) and co-treated with DMSO or 20 μM Parthenolide (PTL). **D** Live-cell imaging of hTERT-RPE1 EGFP-HURP/HaloTag-CENP-A cell blocked in metaphase treated with DMSO or 20 μM Parthenolide (PTL). Inset shows sister K-fibre pair used for kymograph (right panel), arrows indicate the HURP-gaps. **E** Immunofluorescence images of metaphase hTERT-RPE1-EGFP-HURP/HaloTag-CENP-A cells treated with indicated taxol (TX) concentrations for 45 min. Partial Z-projections of 1.0, 0.8, 0.9, 1.1 and 1.3 μm thickness, respectively. Bars show HURP-gap distances. Z-projection = projection of whole spindle. **F** Boxplots of relative HURP intensities in taxol-treated cells as shown in **E** ($N = 4$; $n = 57$, 54, 65, 58 and 67 cells for DMSO, 15, 25, 50 and 100 nM, respectively). Boxplots indicate the 25th and 75th percentiles, the bars are medians, and the whiskers indicate minima and maxima; $P$ = Kruskal–Wallis test and Dunn's multiple comparisons. **G** HURP-gap size distributions in fixed metaphase hTERT-RPE1-eGFP-HURP/HaloTag-CENP-A cells treated with DMSO, or 15 and 25 nM Taxol. ($N = 4$; $n = 272$, 345 and 315 gaps for DMSO, 15 and 25 nM, respectively). Lines = curve fit. Scale bars = 5 μm and 1 μm (kymograph). Source data for all graphs are provided as a Source Data file.

mainly suppress microtubule-ends dynamics, at higher concentrations it stabilises microtubules and induces a GTP-tubulin-like conformation in the microtubule lattice[33–39]. Consistently, kinetochore-tracking and immunofluorescence experiments indicated that increasing taxol doses progressively led to rigid and straight K-fibres with decreasing chromosome movements (Fig. 3E and Supplementary Fig. 2A). As long as kinetochores could still move, HURP stripe intensities increased with increasing taxol concentrations (up to 50 nM), while the size of the HURP-gap shrunk (Fig. 3E–G). HURP levels, however, decreased by 31% at the highest taxol concentration (100 nM) when compared to DMSO/control, when chromosome movements were frozen, and taxol is presumably imposing a GTP-tubulin-like conformation on the microtubule lattice (Fig. 3E, F and Supplementary Fig. 2A). The general HURP decrease was not due to the loss of K-fibres since staining against the spindle checkpoint protein Mad2, a marker of unattached kinetochores[40], revealed few, if any, unattached kinetochores (Supplementary Fig. 2B).

### HURP is excluded from the GTP-cap

To investigate whether the in vivo behaviour of HURP can be explained by the intrinsic properties of the protein or requires other external factors, we expressed and purified recombinant human TagRFP-HURP from insect cells (Supplementary Fig. 3A, B, see "Methods" for details). Mixing this protein with dynamic microtubules assembled from GMPCPP stabilised porcine-tubulin seeds showed that at low nanomolar concentrations TagRFP-HURP randomly bound the microtubule lattice but became enriched at the plus-ends of depolymerising microtubules (Fig. 4A), resembling the accumulation of EGFP-HURP at the leading (depolymerisation-coupled) kinetochore in vivo (Figs. 1 and 2). TagRFP-HURP also dramatically decreased the microtubule catastrophe frequency (from $2.93 \pm 0.33\,h^{-1}$ for the control to $0.20 \pm 0.10\,h^{-1}$ for 50 nM HURP) and increased the rescue frequency (from $282.08 \pm 31.15\,h^{-1}$ for the control to $1107.69 \pm 553.85\,h^{-1}$ for 50 nM HURP), despite exerting little to no effect on both the polymerisation and depolymerisation speeds (Fig. 4B). This rescue factor activity of HURP was consistent with its microtubule-stabilising role in cells[23,24] and our observed correlation between increasing HURP and the directional switching of the leading kinetochore (Fig. 2C–E).

Since End-binding protein 3 (EB3) preferably associates with the GTP-cap of polymerising microtubules[13,14,16–19], we added recombinant EB3-GFP to this minimal system to pinpoint the location of the GTP-cap of in our assay[14,18]. Kymographs showed that on the microtubule plus-ends TagRFP-HURP mirrors the binding pattern of EB3-GFP, which was only associated with growing plus-ends (Fig. 4C). To confirm that recombinant HURP is excluded from the GTP-cap, we averaged the EB3, HURP and tubulin intensity profiles of 123 microtubules on frames where the microtubules were growing. We found that the averaged TagRFP-HURP intensity profile was right-shifted (away from plus-end) by ~150 nm relative to the EB3 peak and the growing microtubule ends. This configuration yielded a gap on single microtubules that contained reduced amounts of both EB3 and HURP (Fig. 4C). In contrast, such gap between microtubule ends and HURP could not be detected on depolymerising microtubule ends (Fig. 4C), nor did we find an equivalent gap at polymerising ends when labelling dynamic single microtubules with the Ska complex, an end tracking complex that is part of the microtubule-kinetochore interface[41–43] (Supplementary Fig. 3C). We conclude that a short HURP-gap, from which EB proteins are mostly excluded, is detectable on polymerising single microtubules in vitro. In living cells, we found an equivalent situation where HURP-gaps are an order of magnitude wider: using hTERT-RPE1 EGFP-HURP/Halo-CENP-A cells expressing EB3-tdTomato we found EB3 closely associated to the trailing kinetochores (bound to growing microtubules), but mostly absent from the HURP-gaps when compared to the signal at kinetochores (Fig. 4D and Supplementary Movie 6). Analysis of HURP and EB3 intensity line profiles revealed an

EB3- and HURP-depleted intermediate zone that could reach up to several microns (Fig. 4E and Supplementary Movie 7).

### HURP preferentially binds the GDP-tubulin microtubule lattices

Our data indicated that HURP is excluded from the GTP-cap and prefers the GDP-tubulin lattice both in vitro and in vivo. This suggested that it may distinguish between the tubulin conformations associated with different nucleotide states. To test this hypothesis, we introduced TagRFP-HURP into flow chambers containing surface-bound porcine brain microtubules that were barcoded such that they contained regions with the slowly hydrolysable nucleotide analogues GMPCPP or GTPγS (reported to mimic GDP + Pi or GTP states[14,15,44]) next to regions with GDP (Fig. 5A). TagRFP-HURP bound the GDP-tubulin lattice fourfold higher than the GMPCPP region and approximately twofold higher than the GTPγS-tubulin region (Fig. 5B; note that both GMPCPP and GTPγS were pre-treated with taxol, but that its presence did not affect the binding of TagRFP-HURP on those microtubules, Supplementary Fig. 4A). Consistently, TagRFP-HURP molecules had twofold higher residency time on the GDP-lattice when compared to the GMPCPP lattice (Supplementary Fig. 4B, C). We conclude that HURP prefers GDP-tubulin microtubule lattices and avoids GTP-like states.

One caveat of this experiment, however, is that these GTP analogues are not necessarily physiological. To substantiate our findings, we also purified wild-type human tubulin and a human tubulin mutant (E254A) incapable of GTP hydrolysis[13,14] (Supplementary Fig. 4D). By extending GMPCPP stabilised porcine-tubulin seeds with either human wild-type- or E254A tubulin (Fig. 5C) we found that TagRFP-HURP avoids the E254A mutant where the GTP state of tubulin is preserved, but not wild-type human tubulin that hydrolyses GTP to GDP (Fig. 5D). We next considered if this nucleotide dependency also holds in living mitotic cells: we transfected hTERT-RPE1 cells with RFP-tagged wild-type or E254 human tubulin. While both human tubulin variants were well incorporated into mitotic microtubule network, E254A tubulin led to longer mitotic spindles, reduced kinetochore oscillation amplitudes and an increase in the EB1 or EB3 signal (Fig. 5E, Supplementary Fig. 4E and Supplementary Movies 8 and 9). This demonstrated that EB1 and EB3 can indeed serve as markers for GTP-tubulin, as had been widely assumed[19]; moreover, it indicated an increase in GTP-tubulin within microtubules[14]. Strikingly, expression of the non-hydrolysing E254A mutant at the same time reduced HURP levels on the mitotic spindle by ~25% (Fig. 5F and Supplementary Movie 9). This shows how increasing the fraction of GTP-tubulin in the spindle is sufficient to displace HURP molecules. Overall, our in vivo and in vitro data reveal how HURP preferentially binds to GDP-tubulin and avoids GTP-tubulin-containing microtubule lattices.

### The mathematical model reproduces observed HURP dynamics

The fact that HURP-gaps also lacked EB proteins raised the possibility of a mixed-nucleotide zone that is neither pure GDP-tubulin nor pure GTP-tubulin, and which would thus fail to accumulate HURP and EBs. To test whether such a configuration would suffice to explain the observed HURP pattern, we constructed a minimal computational model of HURP dynamics on K-fibres. This model captures HURP diffusion, reduced HURP binding on the GTP-cap and the mixed-nucleotide zone, interaction with the RanGTP gradient, and the polymerisation speed of the kinetochore-fibre plus-end (Fig. 6A). The model was then fitted to our experimental lattice light-sheet data (HURP intensity as function of distance from leading or trailing kinetochore over time; see Fig. 2D) using a Bayesian inference framework (see "Methods" for details). This approach allows us to estimate values for model parameters and quantify their uncertainty (shown via posterior distributions) given the observed data (Fig. 6B and Supplementary Table S1).

We next ran forward simulations of the model using the inferred parameter values and found that the model could

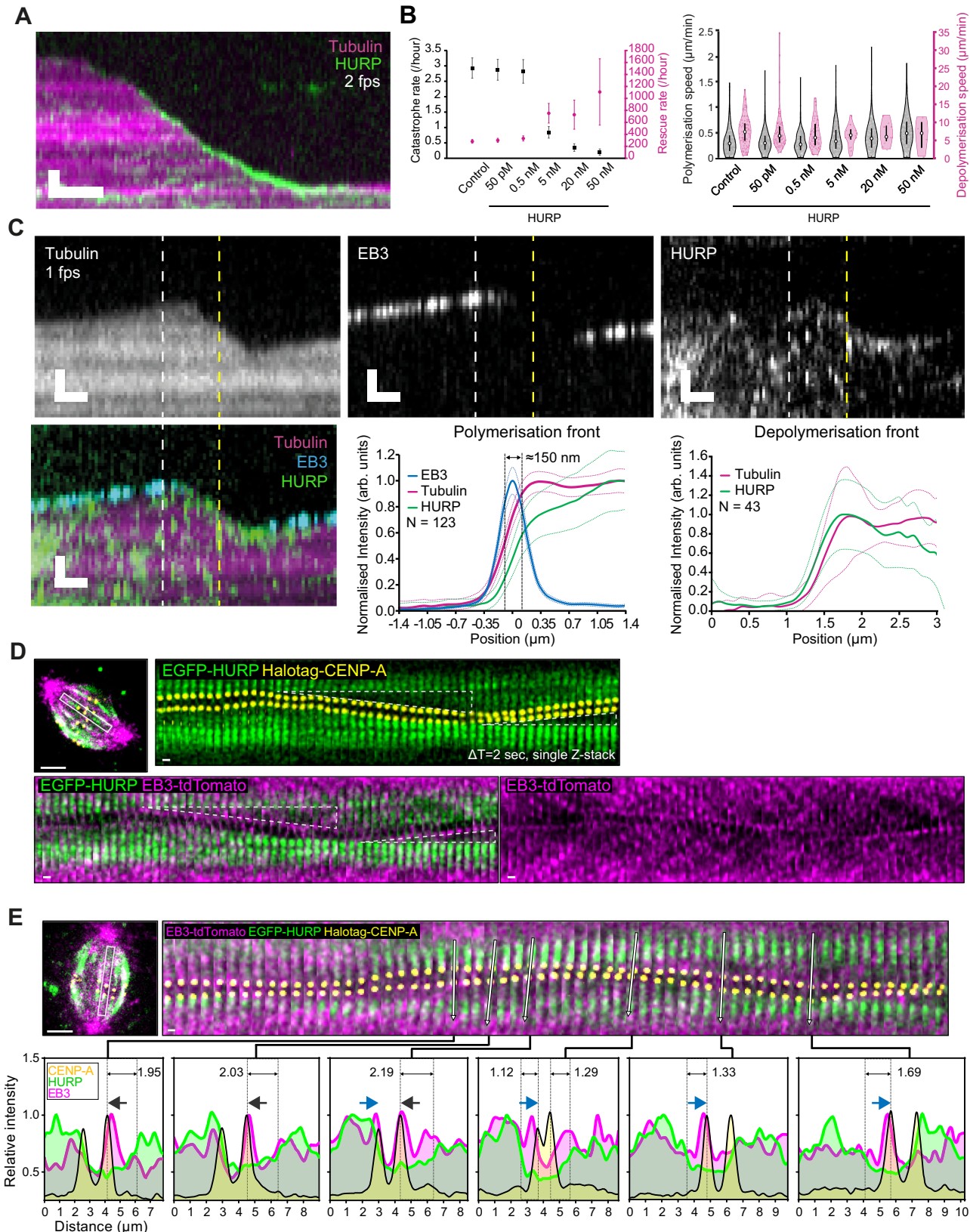

reproduce the experimentally observed HURP dynamics on leading and trailing K-fibres (Fig. 6C). Exploring the sensitivity of these simulations to key model parameters, we saw that the HURP-gap scaled with the size of the mixed-nucleotide zone (estimated as 1.53 μm [0.94, 2.46] μm; see Supplementary Table 1) and speed of K-fibre polymerisation (Fig. 6D). To verify the model, we assessed

model predictions, including the diffusion constant of HURP molecules. When we tracked single HURP particle movement on the microtubule lattice, we obtained a diffusion coefficient, $D = 0.024$ μm²/s, consistent with the posterior distribution for D obtained from fitting the model (Fig. 6E). Furthermore, we tested whether this parametrised model could recapitulate our in vivo

**Fig. 4 | HURP and EB3 are excluded from a micron-wide zone on growing K-fibres. A** Representative kymograph of 5 nM TagRFP-HURP binding to a dynamic microtubule; scale bar = 1 μm vertically, 10 s horizontally. **B** Quantification of the microtubule catastrophe and rescue rate (left) and microtubule polymerisation and depolymerisation rate (right, mean ± standard error) at different TagRFP-HURP concentrations; $N$ = 4, 4, 3, 3, 3 experimental replicates and $n$ = 346, 331, 264, 333, 321 and 181 microtubules analysed for control, 50 pM, 0.5, 5, 20 and 50 nM, respectively. **C** Representative kymographs of 5 nM TagRFP-HURP and 30 nM EB3-GFP binding to a dynamic microtubule (top, scale bar = 1 μm vertically, 10 s horizontally); mean profile of tubulin, EB3 and HURP intensities at the polymerisation and depolymerisation front (bottom, $n$ = 123 and 43 profiles, respectively); white

and yellow vertical dashed lines in the kymographs indicate the representative frame used for quantification at the polymerisation and depolymerisation front, respectively. **D** Representative images of HURP, CENP-A and EB3 signals in the hTERT-RPE1 EGFP-HURP/HaloTag-CENP-A cell line overexpressing EB3-tdTomato (ΔT = 2 s). Triangles highlight HURP-gaps. **E** Representative intensity line profiles of HURP, CENP-A and EB3 signals based on live-cell imaging of hTERT-RPE1 EGFP-HURP/HaloTag-CENP-A overexpressing EB3-tdTomato (ΔT = 2 s). White arrows indicate the selected axis of profiling. Dark grey and blue arrows show the movement direction of kinetochores. Dashed lines indicate HURP-gaps. **D, E** Scale bars = 5 μm and 1 μm (kymograph). Source data for all graphs are provided as a Source Data file.

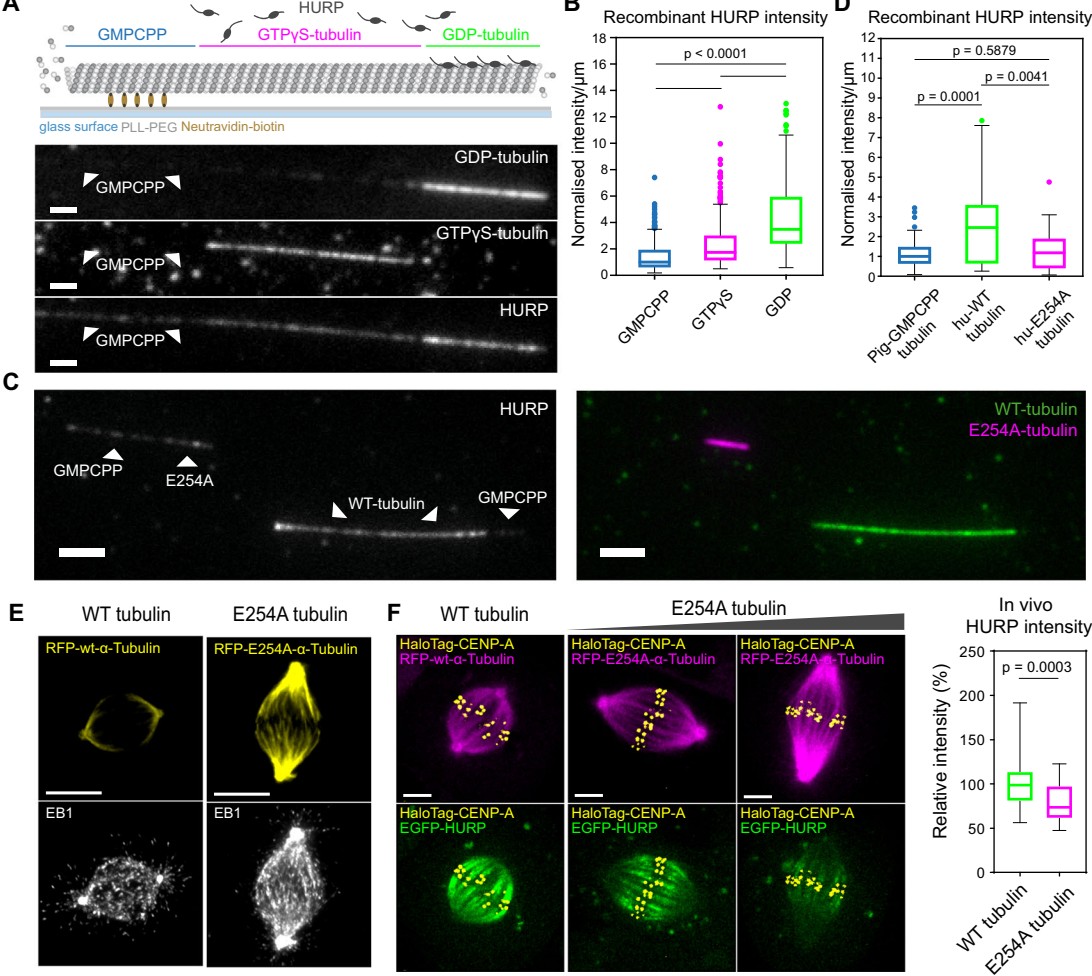

**Fig. 5 | HURP preferentially binds the GDP-lattice of microtubules. A** Schematic of barcoded porcine-tubulin microtubule with the GMPCPP-, GTPγS- and GDP-tubulin sections (top), along with corresponding fluorescence channels of GDP- (HiLyte647), GTPγS- (HiLyte488) tubulin sections (unlabelled GMPCPP-tubulin, see "Methods") and TagRFP-HURP (bottom); scale bars = 1.5 μm. **B** Boxplots of the TagRFP-HURP intensity/μm normalised to its median intensity on GMPCPP microtubules ($N$ = 3 flow chambers; $n$ = 171 (GMPCPP), 300 (GTPγS) and 238 (GDP); $P$ = Kruskal–Wallis test). **C** Representative image of porcine-tubulin GMPCPP seeds extended with either human WT- (green) or E254A tubulin (magenta), labelled with HiLyte488 and HiLyte647 porcine brain tubulin (1:7 labelled:unlabelled ratio) respectively (right panel) and incubated with TagRFP-HURP (left panel); scale bars = 3 μm. **D** Boxplots of HURP intensity/μm normalised to its median intensity on

GMPCPP seeds ($N$ = 3 flow chambers; $n$ = 121 (GMPCPP), 43 (WT) and 87 (E254A tubulin); $P$ = Kruskal–Wallis test). **E** Immunofluorescence images of metaphase hTERT-RPE1 cells transduced either with RFP-wt-α-tubulin or RFP-E254A-α-tubulin and stained for EB1. Scale bars = 5 μm. **F** Live-cell imaging of hTERT-RPE1 EGFP-HURP/HaloTag-CENP-A cells transduced either with RFP-wt-α-tubulin or RFP-E254A-α-tubulin mutant (left), Z-projections of 5 × 0.5 μm, Scale bar = 5 μm; and boxplots of in vivo relative HURP intensities (right). $N$ = 2; $n$ = 38 and 42 cells. Boxplots indicate the 25th and 75th percentiles, the bars are medians, and the whiskers indicate values within 1.5 times the interquartile range in **B, D**, and minima and maxima in **F**; $P$ = two-sided Mann–Whitney test. Source data for all graphs are provided as a Source Data file.

FRAP measurements. We, therefore, simulated a FRAP experiment (Fig. 6F) by using an initial condition featuring HURP bleached from a region 1.5 μm away from the chromosomes and found that the HURP distribution recovered over a timescale (~10 s) similar to

that measured in our in vivo FRAP experiments (see Fig. 3A, B). We conclude that a micron-sized mixed-nucleotide zone is sufficient to explain the appearance of the observed EB/HURP-negative gap on the polymerising K-fibre.

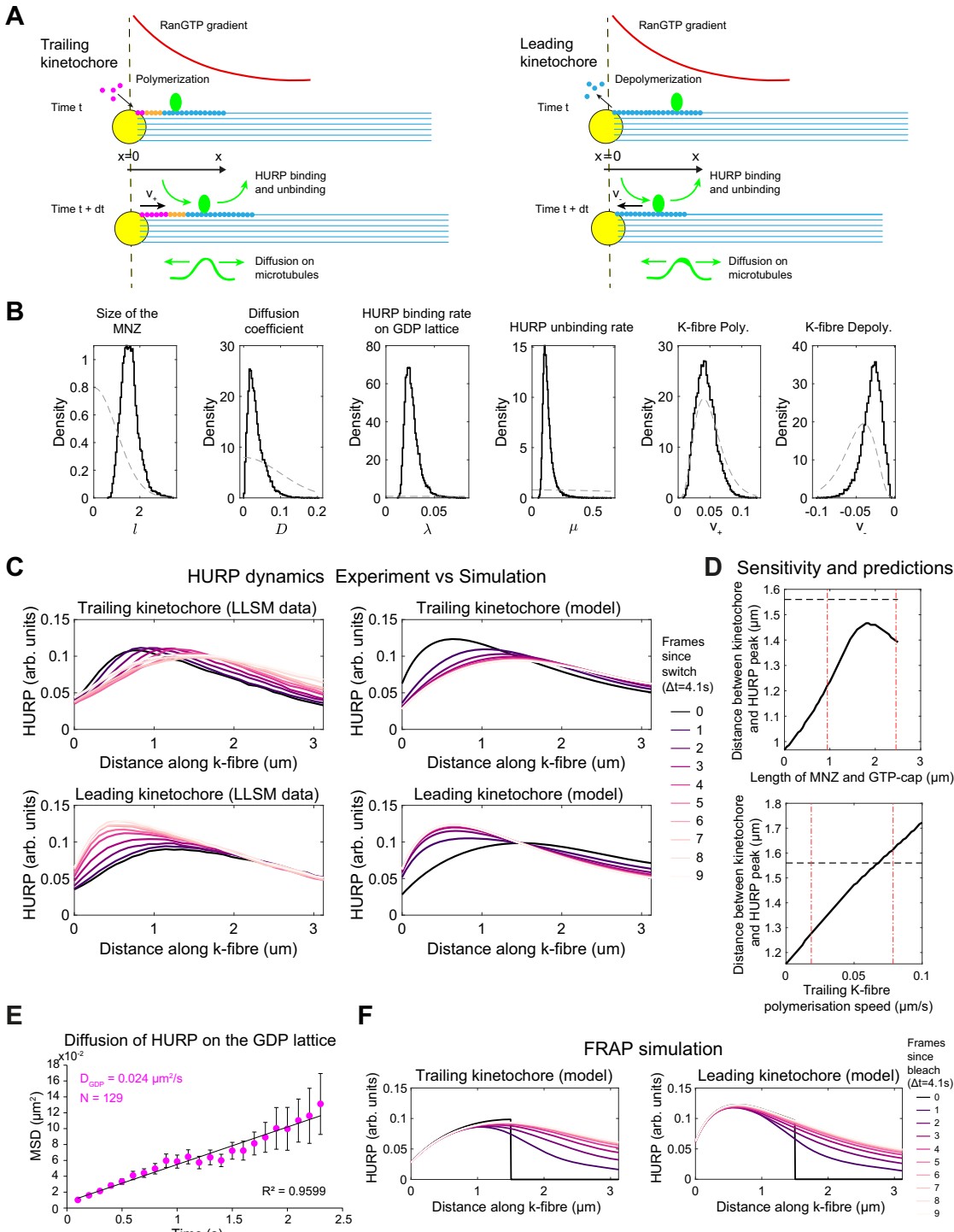

**Fig. 6 | A minimal computational simulation of spatiotemporal dynamics of HURP recapitulates HURP-gaps. A** Schematic showing the key components of the computational model of HURP dynamics on K-fibres, including diffusion, binding and unbinding with rates dependent on the RanGTP gradient (red) and tubulin conformation (GTP–magenta, mixed-nucleotide zone–orange, GDP–blue), and polymerisation/depolymerisation at the kinetochore resulting in the advection of HURP. **B** Estimated distributions of key model parameters (histograms of posterior marginal distributions) based on data from Fig. 2C (see "Methods"). Grey dashed lines indicate prior distributions (i.e., before learning from the data). **C** Spatiotemporal dynamics of HURP on K-fibres from in vivo data (as in Fig. 2C) and computational model predictions using fitted parameters (median of each posterior marginal distribution). **D** Sensitivity of the model to changes in the length of GTP-cap and mixed-nucleotide zone (top), and rate of polymerisation of the trailing K-fibre (bottom) showing how these affect the size of the HURP-gap. Other parameters remain fixed at posterior median values. Vertical dashed red lines indicate the 95% credible region. A horizontal dashed black line shows the measured HURP-gap from lattice light-sheet data. **E** Plots of the mean squared displacement (MSD) of HURP over time measured from in vitro data (mean ± SEM; $N = 1$, $n = 129$ traces). **F** Simulated FRAP experiment using parameters estimated from data (median of each posterior marginal distribution) and assuming initially HURP was bleached from a region beyond 1.5 μm from the kinetochore (zero HURP assumed initially in this region). Source data for all graphs are provided as a Source Data file.

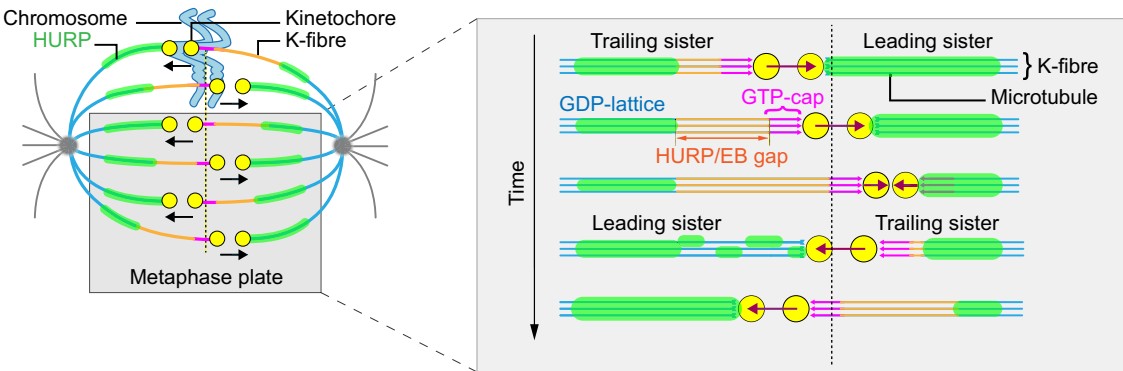

**Fig. 7 | Proposed model of the mixed-nucleotide zone on K-fibres.** Based on our experimental data, we propose that K-fibres contain four distinct regions: (1) A GTP-cap (magenta) close to the trailing kinetochore, characterised by an accumulation of EB proteins; (2) a micron-sized dynamic mixed-nucleotide zone (orange) on growing K-fibres that contains neither EB proteins nor HURP; (3) the GDP microtubule lattice (blue) characterised by HURP stripes; (4) a pole-proximal region that lacks HURP due to the absence of RanGTP. Note that as soon as a directional switch occurs, HURP starts to accumulate on the leading K-fibre that previously contained HURP-gap, and an equivalent gap starts to form on the opposite, growing K-fibre.

## Discussion

Based on our data, we postulate that the classical model of the K-fibres on the mitotic spindle, composed of a GTP-tubulin cap and a GDP-tubulin microtubule lattice should incorporate a micron-sized mixed-nucleotide zone on the newly polymerising K-fibre (Fig. 7). Indeed, our results reveal the presence of four distinct regions on growing K-fibres: (1) a EB3-positive/HURP-negative GTP-cap abutting the kinetochore, followed by (2) an EB3/HURP-negative zone that can extend for several microns, (3) a HURP-positive GDP-bound lattice (the HURP stripes) and (4) an EB3/HURP-negative pole-proximal zone of GDP-lattice. Our in vitro reconstitution data and the fact that E254A tubulin displaced HURP on the entire spindle, indicate that the HURP-gap is a direct consequence of HURP preferentially binding GDP-tubulin microtubule lattices. In contrast, we could exclude the detyrosination/tyrosination cycle as a cause for the HURP-gap. Moreover, we note that high doses of taxol, which are associated with high doses of tubulin acetylation[45-47], led to a reduction of HURP levels on the spindle, arguing against HURP recognising acetylated tubulin. Thus, we could exclude the two post-translational tubulin modifications that could have best explained the observed HURP pattern. Given that the HURP-gap represents a region that neither recruits a GTP-tubulin (EBs) nor a GDP-tubulin (HURP) marker, we conclude that it must represent a mixed-nucleotide zone. Based on our mathematical modelling, we predict that the size of this mixed-nucleotide zone is slightly smaller, but proportional to the HURP-gap size (Fig. 6D).

Our in vivo data show that the transition into and from this mixed-nucleotide zone is sharp for both EB proteins and HURP, suggesting a cooperative binding process for both proteins that requires a specific threshold of GTP- or GDP-tubulin. Consistently, EB proteins bind to microtubules at the interface of four tubulin subunits[17,48] and require GTP-tubulin on adjacent tubulin subunits to bind to microtubules[18]. We speculate that HURP may recognise the GDP-tubulin conformation at an interface located between adjacent tubulin subunits via a similar mechanism. Future structural analysis will, however, be necessary to dissect the mechanism that confers HURPs preference for GDP-tubulin. Because our in vitro data also show that HURP is a potent microtubule-rescue factor it is tempting to speculate that HURP stripes establish global boundaries of microtubule depolymerisation lifetime within the spindle. More generally, our work reveals HURP is a mitotic spindle-associated protein that specifically recognises GDP-tubulin. Since Doublecortin (DCX) and Tau have been shown to also avoid the GTP-cap in post-mitotic neuronal cells[49,50] there is growing support for a new class of growing microtubule tip-avoidance factor.

Our in vivo data reveal that the HURP-gap reflects an underlying mixed-nucleotide zone on the growing K-fibre which can reach several microns in cells, much larger than the one detected in vitro within a single microtubule (~150 nanometres). This difference could be either due to other microtubule-associated proteins affecting the GTP hydrolysis rate in K-fibres, and/or suggests that parallel bundling of microtubules imparts alterations to the nucleotide and/or structural transitions within the lattices. Low nanomolar concentrations of taxol, which reduces the tubulin-incorporation rate, led to smaller HURP-gaps, implying its size can serve as an indicator for the GTP-tubulin-incorporation/hydrolysis rate. More generally, the persistence of the HURP-gap for 45 s and more, indicates that the GTP-tubulin hydrolysis rate in growing K-fibres is very slow. This suggests that GTP hydrolysis within the K-fibre associated with the trailing kinetochore is unlikely to dictate sister-kinetochore directional switches. This is consistent with previous observations, showing that such switches are mostly initiated by the leading kinetochore[51]. In contrast, once a directional switch occurs, HURP-gaps are immediately replenished by HURP, implying a very rapid wave of GTP hydrolysis along the microtubule lattice once a K-fibre starts shrinking. Overall, the dynamics of HURP indicate that the K-fibre lattice is not a homogeneous structure but rather that its nucleotide content is highly dynamic in nature. This present work defines the existence of a new mixed-nucleotide zone within the mitotic spindle and raises the possibility for equivalent regions in other microtubule-based assemblies.

## Methods

### Cell culture, cell lines and treatments

hTERT-RPE1 (non-transformed immortalised human retina pigment epithelial cells) (ATCC; CRL-4000), hTERT-RPE1 Centrin1-GFP/GFP-CENP-A (kind gift from A. Khodjakov), HCT-116 (kind gift from P. Nowak-Sliwinska) (ATCC; CCL-247) and hTERT-RPE1 EB3-GFP (kind gift from A. Straube[18]) cells were cultured in high glucose DMEM (Thermofisher; 41965) supplemented with 10% FBS (LabForce; S1810) and 1% penicillin/streptomycin (Thermofisher; 15140). The medium for hTERT-RPE1 EGFP-HURP/HaloTag-CENP-A, HeLa EGFP-HURP/HaloTag-CENP-A and hTERT-RPE1 EGFP-HURP/HaloTag-CENP-A/EB3-tdTomato cells was supplemented with 600 μg/mL G418 (InvivoGen; ant-gn-5). ECRF24 cells[52] (Human immortalised endothelial cells) (kind gift from P. Nowak-Sliwinska and were initially generously donated by Prof. AW Griffioen (Angiogenesis Laboratory, UMC Amsterdam, The Netherlands)) were cultured in flasks coated with 0.2% gelatine and grown in 1:1 DMEM-RPMI-1640 (Thermofisher; 41965, 21875) supplemented with 10% FBS and 1% penicillin/streptomycin.

To generate the HeLa EGFP-HURP cell line, HeLa cells (ATCC; CCL-2) were co-transfected with the pCMV-Cas9-RFP plasmid (Sigma-Aldrich, encoding for both Cas9 gene and the guide RNA targeting the first exon of the human *DLGAP5* gene) along with the EGFP-HURP repair template cloned into the pUC57-kan plasmids (Genewiz)[24]. Positive EGFP clones were then selected, and the genomic DNA was sequenced.

To generate hTERT-RPE1 EGFP-HURP/HaloTag-CENP-A or HeLa EGFP-HURP/HaloTag-CENP-A cells, hTERT-RPE1 EGFP-HURP or HeLa EGFP-HURP cells[24] were transfected with pHTN HaloTag CENP-A, selected with G418, and FACS sorted twice for HaloTag-CENP-A-positive cells. To mark kinetochores, cell lines expressing HaloTag-CENP-A were incubated for 30 min with 20 nM 646- or 549-Janelia Fluor HaloTag ligand (Promega; GA1120, GA1110) 16 h prior to any experiment. For lattice light-sheet imaging, cells were treated with 100 μM TMR (Promega; G8252) for 15 min prior to imaging at least 30 min later. The following drugs were used in this study: cells were arrested in metaphase with 10 μM MG132 (Sigma-Aldrich; C2211) for 45 min and treated with indicated concentrations of taxol (Sigma-Aldrich; T7402) 45 min before acquisition or fixation. 1 μg/mL nocodazole (Sigma-Aldrich; M1404) was used to create unattached kinetochores. To prevent tubulin detyrosination, cells were incubated with 20 μM Parthenolide (Sigma-Aldrich; P0667) for 45 min, followed by a 45 min treatment with 10 μM MG132 before imaging or cell harvesting.

## Live-cell imaging

Cells were seeded in polymer coverslip two-, four- or eight-well μ-Slide ibidi chambers (Ibidi; 80286, 80426, 80826). 4 h prior live-cell imaging, the medium was replaced by Leibovitz L-15 (Thermofisher; 21083) supplemented with 10% FBS and 1% penicillin/streptomycin. The chambers were acclimatised in a 37 °C chamber before imaging. The acquisition was performed using an EC Plan Neofluar 100X (NA 1.3) oil objective on a Zeiss Cell Observer.Z1 spinning disk microscope (Nipkow Disk) equipped with a 37 °C chamber and a CSU X1 automatic Yokogowa spinning disk head. 512 ×512 pixels images were acquired with an Evolve EM512 camera (Photometrics) using Visiview 4.00.10 software. For the in vivo visualisation and quantification of EGFP- HURP-gaps, metaphase hTERT-RPE1 EGFP-HURP/HaloTag-CENP-A or HeLa EGFP-HURP/HaloTag-CENP-A cells were imaged every 5 s for 2 or 3 min with 5 Z-stacks of 0.5 μm spacing. For co-visualisation of EGFP-HURP and EB3-tdTomato on K-fibres or visualisation of EB3-GFP in presence of wt- or E254A-α-tubulin, cells were imaged either on a single focal plane or 3 Z-stacks of 0.5 μm spacing, every 2 s for 2 min. Lattice light-sheet imaging was performed with a lattice light-sheet microscope from Intelligent Imaging Innovations (3i; https://www.intelligent-imaging.com). Cells were seeded on 5 mm radius glass coverslips one day before transfer of coverslip to the LLSM bath filled with $CO_2$-independent L-15 medium, where live imaging took place. 3D time-lapse images of hTERT-RPE1 EGFP-HURP/HaloTag-CENP-A cell line were acquired at 488-nm and 560-nm channels using 1% and 4% laser power, respectively, 35 ms exposure time/z-plane, 90 z-planes, 308 nm z-step, resulting in 4.14 s/z-stack time (frame). Acquired movies were de-skewed and cropped in XYZ and time, using Slidebook software. Cropped movies were saved as OME-TIFF files in ImageJ.

## Immunofluorescence

For visualisation of EGFP-HURP or endogenous HURP, cells were grown on acid-etched glass coverslips, permeabilized with 0.5 % Triton X-100 in 37 °C pre-warmed Microtubule Stabilising Buffer (MTSB buffer–80 mM KOH-PIPES (pH = 6.8), 1 mM MgCl₂, 5 mM EGTA) for 30 s and fixed with 0.5% Glutaraldehyde (Sigma-Aldrich; G6403) in MTSB buffer for 10 min at RT. After fixation, glutaraldehyde was quenched by 0.1% NaBH₄ in PBS for 7 min at RT. For detection of Mad2, cells were fixed 10 min with 4% formaldehyde (Sigma-Aldrich; 47608) in 20 mM

KOH-PIPES (pH = 6.8), 10 mM EGTA, 1 mM MgCl₂, 0.2% Triton X-100. For visualisation of EB1 cells were fixed in −20 °C methanol (Sigma-Aldrich; 32213) for 6 min, rinsed with PBS. After fixation all cells were rinsed with PBS and blocked for 30 min in PBS + 3% BSA (PAN-Biotech; P06-1391500). All primary and secondary antibodies were diluted in PBS + 3% BSA. The following primary antibodies were used: rabbit anti-GFP (1:500[53]), chicken anti-GFP (1:1000; Thermofisher; A10262), rabbit anti-HURP (1:500; kind gift from E. Nigg[23]), recombinant human anti-α-tubulin (1:300[54]), rabbit anti-α-tubulin (1:1000; Abcam; ab18251), mouse anti-Hec1 (1:700; Abcam; ab3613), rabbit anti-Mad2 (1:1000; Bethyl; A300-301A), and mouse anti-EB1 (1:200; BD biosciences; 610535). For cells expressing EGFP-HURP, anti-GFP and anti-α-tubulin antibodies were added for 1 h, whereas for detection of endogenous HURP anti-HURP, anti-Hec1 and anti-α-tubulin antibodies were added for 2 h. After PBS washes, coverslips were incubated 1 h with appropriate Alexa Fluor–conjugated secondary antibodies (1:400; Invitrogen; A-11008, A-11036, A-31573, A-21202, A-11031, A-31571, A-21090, A-21445, A-11039). Immunolabelled cells were washed PBS before mounting the coverslips with VECTASHIELD with/without DAPI (Vector Laboratories; H-1200, H-1000). Immunofluorescence images were acquired on an Olympus DeltaVision wide-field microscope (GE Healthcare) equipped with a DAPI/FITC/TRITC/Cy5 filter set (Chroma Technology Corp.) and a Coolsnap HQ2 CCD camera (Roper Scientific) running Softworx 6.5.2 (GE Healthcare). 3D images were deconvolved using Softworx 6.5.2 (GE Healthcare). To visualise EGFP-HURP or endogenous HURP, mitotic spindles were imaged using a 100×1.4 NA objective in 0.1 μm Z-stacks; for all other antibodies, mitotic spindles were imaged with a 100×1.4 NA or a 60X NA 1.4 objective in 0.2 μm Z-stacks. Alternatively, endogenous HURP and HURP-gaps were visualised with an EC Plan Neofluar 100X (NA 1.3 Oil Ph3) objective in 0.1 μm Z-stacks on a spinning disk microscope (Nipkow Disk) Zeiss Cell Observer.Z1 equipped with a HXP120 fluorescence wide-field visualisation lamp and with a CSU X1 automatic Yokogawa spinning disk head. 512 ×512 pixel images were acquired with an Evolve EM512 camera and Visiview 4.00.10. Spinning disk 3D images were deconvolved using Huygens Essential v.21.10 software.

## Fluorescence recovery after photobleaching

Cells were seeded in a glass bottom four-well μ-Slide 4 Ibidi chamber (Ibidi; 80427). Cells were incubated in imaging medium 4 h prior FRAP experiment. Metaphase cells were selected based on brightfield contrast and single focal planes of 140 nm pixel size were acquired using a 60× (NA 1.4) CFI Plan Apochromat oil objective on a Nikon A1r point scanning confocal microscope and NIS Elements 4.30.02 software. Single K-fibres were visualised based on EGFP-HURP signal and targeted with a circular ROI of 10 pixels (1.3 μm) in diameter before photobleaching with two 500-ms pulses of 80% 405 nm laser power. After bleaching, the cells were acquired on a single focal plane every 2 s for 2 min. FRAP experiments were analysed using a custom-made application written in Matlab 2021a (MathWorks, Natick, MA, USA), available at https://github.com/Bioimaging/FRAP. Briefly, if present, time-lapse images drift was corrected using rigid registration. Three classes of the region of interest were manually positioned to report the bleached area, the whole mitotic spindle, and the background area fluorescent intensities. The FRAP signal was computed using a double normalisation procedure as followed:

$$FRAP(t) = \frac{Bleach(t) - Back(t)}{Bleach_{pre} - Back_{pre}} \cdot \frac{Ref_{pre} - Back_{pre}}{Ref(t) - Back(t)} \quad (1)$$

where Bleach(t) is the spatial average intensity in the bleached region, $Bleach_{pre}$ is the average intensity before bleaching, Ref(t) is the spatial average intensity in the whole mitotic spindle, $Ref_{pre}$ is the average intensity before bleaching, Back(t) is the spatial average intensity in the background region, and $Back_{pre}$ is the average intensity before

bleaching. Such normalisation accounts for the photobleaching induced by the imaging laser.

## Image processing and analysis

To quantify the maximum distance and live duration of HURP-gaps, movies were analysed using Imaris 7.7 software (BitPlane). A maximum intensity projection yielded 2D movies; potential translational and rotational drift were corrected using surfaces tool based on whole EGFP-HURP signal. Kinetochores were automatically or manually detected using spots tool based on the CENP-A signal. HURP-gap lengths were measured using the Measurement points tool as the distance between the CENP-A centre and the edge of the EGFP-HURP signal on growing K-fibres. To visualise HURP on sister K-fibre pairs and quantify HURP-gap distances in fixed cells, deconvolved images were visualised in 3D using Imaris 7.7. Only cells displaying mitotic spindles parallel to x axis were selected, and sister K-fibre pairs were identified based on α-tubulin and CENP-A signals. HURP-gaps lengths were measured as in live cells. EGFP-HURP intensities on mitotic spindles of immunolabelled taxol-treated hTERT-RPE1 EGFP-HURP/HaloTag-CENP-A cells were quantified using ImageJ/Fiji software. EGFP-HURP intensities were averaged from three different Z-slices localised in the upper, middle, and lower parts of the spindle. In each slide, we measured the integrated density (IntDen) of EGFP-HURP per half spindle inside an ROI of $25 \times 120$ pixels ($1.620\,\mu m \times 7.776\,\mu m$) enclosing K-fibres close to kinetochores. EGFP-HURP background intensity was subtracted after performing the same measurement in a ROI in a cytosolic region close to the spindle pole. EGFP-HURP intensities in live-cell imaging movies of hTERT-RPE1 EGFP-HURP/HaloTag-CENP-A cells transfected with WT- or E254A-α-tubulin were quantified using the same procedure, but this time on the summed Z-projection of 5 $\times 0.5\,\mu m$ stacks and an ROI of $50 \times 100$ pixels ($8\,\mu m \times 16\,\mu m$) enclosing K-fibres (RFP signal) of the kinetochore region (CENP-A signal).

## Kinetochore-tracking assay

hTERT-RPE1 Centrin1-GFP/GFP-CENP-A cells were seeded in a four-well μ-Slide Ibidi chambers, treated with $10\,\mu M$ of MG132 for 45 min followed by indicated taxol doses for an additional 45 min before acquisition. Metaphase-arrested cells were imaged in the GFP channel with $2 \times 2$ binning, every 7.5 s for 5 min and $15\,\mu m$ Z-stacks were acquired in $0.5\,\mu m$ steps. All movies were acquired using a $100\times$ NA 1.4 objective on an Olympus DeltaVision wide-field microscope equipped with an eGFP/RFP filter set (Chroma), and a Coolsnap HQ2 CCD camera running Softworx. 3D movies were deconvolved and cropped using Softworx. Deconvolved movies were analysed using an automated kinetochore-tracking software (KiT) written in MATLAB 2013b[11]. The latest version of the code is available at https://github.com/cmcb-warwick/KiT. Briefly, the frame-to-frame displacement of sister kinetochores and their relative distance from the centre of the metaphase plate were analysed, and an autocorrelation function was used to quantify the regularity of the sister-kinetochore oscillations along the spindle axis. For lattice light-sheet imaging, kinetochore tracking and sister pairing were performed using the CENP-A channel with a version of the kinetochore-tracking software (KiT) adapted for lattice light sheet data[55]. Line profiles were measured in the direction of the inter-kinetochore vector (smoothed in time with a 5-point stencil) to quantify the spatial distribution of HURP on K-fibres. Line profiles extended $3\,\mu m$ along the K-fibre and linear interpolation between pixels was applied where necessary. K-fibre polymerisation states were annotated automatically via Bayesian inference using a model of kinetochore dynamics in metaphase[27]. This model takes the form of a hidden Markov model with forces due to K-fibre polymerisation, polar ejection force, and the chromatin spring joining sister chromatids. Hamiltonian Monte Carlo is used to sample model parameters with the likelihood evaluated via the forward algorithm, and backward sampling used to obtain the polymerisation states[56].

## Plasmids, lentivirus production and transfections

To generate the pHTN HaloTag CENP-A plasmid, a synthetic cDNA encoding human CENP-A was subcloned into the pHTN HaloTag CMV-neo vector (Promega) using EcoRI and XbaI sites. The EB3-tdTomato plasmid (kind gift from E. Dent; Addgene 50708) was used to label GTP-tubulin caps. For transfections, the culture medium was exchanged with antibiotic-free MEM (Thermofisher; 41090) supplemented with 10% FBS prior to transfection. Cells were transfected with X-tremeGENE 9 DNA Transfection Reagent (Roche; 6365779001) at a ratio of 3:1 μl/μg DNA. The TagRFP-HURP construct used for all in vitro experiments was ordered from GeneArt using the HURP gene sequence for isoform 1 (Uniprot Q15398), optimised for Sf9 insect cell expression (Spodoptera Frugiperda). The construct starts with a N-terminal 6× Histidines tag, followed by a linker containing a Pre-scission protease recognition sequence (SGVLFQGP), the TagRFP sequence, a linker comprising of five Glycines, and ends with the HURP sequence. The α-tubulin E254A mutant was generated by site-directed mutagenesis (Stratagene) using the pmRFP-α-tubulin IRES puro 2b plasmid as a template (kind gift from D. Gerlich; Addgene 21041). The WT- and E254 RFP-α-tubulin sequences were introduced into the pCDH-CMV hGLuc (kind gift from M. Strubin) vector using the SpeI and NotI restriction sites. For the production of VSV-G pseudotyped recombinant lentiviruses, $4.5 \times 10^6$ HEK 293 T/17 cells (ATCC; CRL-11268) were seeded into a 10-cm dish and transiently transfected for 16 h by calcium phosphate precipitation with $10\,\mu g$ of packaging plasmid psPAX2, $5\,\mu g$ of envelope plasmid pMD2G (both kind gifts from D. Trono; Addgene 12260 and 12259) and with $15\,\mu g$ of pCDH-CMV-RFP-α-tubulin (wt) or pCDH-CMV-RFP-α-tubulin E254A. Supernatants containing recombinant viruses were collected 48 h post transfection and filtered through PVDF 0.45-μm filters (Merck-Millipore). For lentiviral transduction, $2 \times 10^4$ RPE1 EGFP-HURP/Halo-CENP-A cells/well were seeded in 4-well μ-Slide ibidi chambers. The day after, cells were incubated with different volumes of virus supernatants, washed with PBS 24 h later, and incubated with Janelia Fluor 646 HaloTag ligand before being placed in fresh medium for an additional 24 h before imaging.

## Protein purification

6xHis-TagRFP-HURP was expressed in Sf9 insect cells. Pellets from 2 L cultures were resuspended in Lysis buffer (50 mM HEPES pH 7.5, 200 mM NaCl supplemented with 1% Igepal, 5% glycerol, 10 mM MgCl$_2$ and DnaseI) and the lysate homogenised with 10 strokes, supplemented with 5 mM DTT (Roche; 10 708 987 001), 1% PMSF (Roche; 11359061001) and 1% SERVA protease inhibitor mix (SERVA; 39107.02), and incubated on ice for 20 min. The lysate was sonicated and centrifuged at $>40,000 \times g$ for 20 min at 4 °C, and the supernatants were loaded onto a SP FF 5 ml column (Cytiva; 17515701) equilibrated in Lysis buffer. The column was washed with Lysis buffer, followed by Buffer A (50 mM HEPES pH 7.5, 240 mM NaCl, 5 mM DTT) and eluted via a 20 CV gradient from 5 to 100% of Buffer B (50 mM HEPES pH 7.5, 1 M NaCl, 5 mM DTT). The eluted fractions were pooled and concentrated into a 30 kDa MWCO spin column (Amicon Ultra-15, Merck-Millipore; UFC903024) at $3240 \times g$ for 40 min at 4 °C, diluted sevenfold using Buffer A and loaded onto two serially connected 1 ml Q sepharose FF columns (Cytiva; 17505301) equilibrated in Buffer C (50 mM HEPES pH 7.5, 200 mM NaCl, 10 mM Imidazole, 1 mM DTT and 0.05% Tween20). These were washed with 8 CV of Buffer C, then both flowthrough and washes were collected, pooled and concentrated as above to exchange in Buffer C. The sample was loaded onto a 5 ml HisTrap FF (Cytiva; 17525501) equilibrated in Buffer C. The column was washed with 8 CV of Buffer C and eluted with a 10 CV gradient to 100% of Buffer D (50 mM HEPES pH 7.5, 200 mM NaCl, 500 mM Imidazole, 1 mM DTT and 0.05% Tween20). The isolated fractions were pooled and concentrated as above and subsequently injected into a Superose 6 Increase gel filtration column (Cytiva; 29091596) equilibrated and run

in Buffer E (50 mM HEPES pH 7.5, 300 mM NaCl, 1 mM DTT). Finally, fractions were pooled, aliquoted and flash frozen and stored in liquid nitrogen. The human tubulin wt- and E254A constructs were a kind gift by T. Surrey and were expressed in Sf9 insect cells and purified as described[14]. Briefly, pellets were resuspended in 20 ml each of Lysis buffer (80 mM PIPES pH = 7.2, 1 mM EGTA, 6 mM MgCl₂, 100 mM KCl, 50 mM Imidazole, 2 mM GTP (Jena Bioscience; NU-1012-10G) and 1 mM DTT, with added 2 complete anti-protease tablet (Roche, 04 693 116 001) and 1:200 DnaseI (10 µg/ml)); the lysate was homogenised with 60 strokes, diluted threefold in Dilution buffer (80 mM PIPES pH = 7.2, 1 mM EGTA, 6 mM MgCl₂, 50 mM Imidazole, 2 mM GTP and 1 mM DTT) and spun down in a T865 rotor at 39500 rpm for 1 h at 4 °C, before loading the supernatant on a 5 ml HisTrap FF (Cytiva; 17525501), equilibrated in lysis buffer. The column was washed sequentially in 10 CV each of Lysis buffer, Wash buffer 1 (80 mM PIPES pH = 7.2, 1 mM EGTA, 11 mM MgCl₂, 2 mM GTP, 5 mM ATP and 1 mM DTT), Wash buffer 2 (80 mM PIPES pH = 7.2, 1 mM EGTA, 5 mM MgCl₂, 0.1% Tween20, 1% w/v glycerol, 2 mM GTP and 1 mM DTT) and finally Lysis buffer again as above. The protein was eluted with Elution buffer (80 mM PIPES pH = 7.2, 1 mM EGTA, 5 mM MgCl₂, 500 mM Imidazole, 2 mM GTP and 1 mM DTT) and further diluted sixfold in Strep Binding buffer (80 mM PIPES pH = 7.2, 1 mM EGTA, 5 mM MgCl₂, 2 mM GTP and 1 mM DTT) and loaded onto a 1 ml HiTrap SP FF column serially connected to a 5 ml StrepTrap FF column (Cytiva; 28907548). The columns were washed in Strep Binding buffer, the HiTrap detached and the StrepTrap further washed for 10 CV. The sample was eluted using Strep Elution buffer (80 mM PIPES pH = 7.2, 1 mM EGTA, 4 mM MgCl₂, 50 mM Imidazole, 2.5 mM Desthiobiotin, 2 mM GTP and 1 mM DTT) and supplemented with 50 µl of TEV protease (NEB; P8112S) for overnight cleavage of the Strep tag at 4 °C. Following on, the sample was clarified at 61000 rpm in a TLA100.3 rotor for 10 min at 4 °C and then loaded onto a 1 ml HiTrap SP FF (Cytiva; 17505401), collecting the flowthrough. Finally, the sample was concentrated in an Amicon 30 KDa MWCO, buffer exchanged in a HiPrep 26/10 desalting column (Cytiva; 17508701) in BRB80 (80 mM K-PIPES pH 6.8, 1 mM MgCl₂, 1 mM EGTA supplemented with 0.2 mM GTP, aliquoted, flash frozen and stored in liquid nitrogen. The EB3-GFP purified protein was kindly gifted by A. Straube. Finally, the EGFP-Ska complex was expressed and purified as described[57].

## Imaging HURP in vitro

The in vitro microtubule assay was performed as described[57]. In brief, microtubules seeds stabilised with 1 mM GMPCPP (Jena Bioscience; NU-405S) and 2 µM taxol (Merck; T7402) were adhered to the coverslip surface of a flow chamber via biotin–streptavidin link. After 5 min incubation, the chamber was perfused with the assay mix, comprising 9 µM free tubulin (1:25 unlabelled:labelled ratio using either Hilyte647 or Hilyte488 (Cytoskeleton; TL670M and TL488M respectively)) in BRB80 buffer (80 mM K-PIPES pH 6.8, 1 mM MgCl₂, 1 mM EGTA) supplemented with 70 mM KCl and Oxygen scavengers (0.4 mg/ml glucose oxidase (Sigma-Aldrich; G7141-50KU), 0.2 mg/ml catalase (Merck; C1345-10G), 50 mM glucose) and the protein of interest. This allowed us to study the binding and unbinding of the protein of interest on dynamic microtubules. Barcoded microtubules were created by adding 16 µM of pig brain tubulin (1:15 labelled:unlabelled) to stabilised GMPCPP seeds (final ratio of 1:1 v:v), in the presence of 1 mM GTPγS (Jena Bioscience; NU-412-2), and incubating the solution at 37 °C for 1 h. The mix was diluted fivefold in BRB80 supplemented with 2 µM taxol (BRB80 + Tx), spun in an airfuge at 20 psi for 10 min at room temperature, before resuspending the pellet in BRB80 + Tx. For the human tubulin experiments, 16 µM of wt-tubulin or 6 µM of E254A tubulin were added to GMPCPP seeds (final ratio of 5:1 v:v), in the presence of 1 mM GTP. Chambers were imaged using TIRF built on an Olympus IX81 inverted microscope equipped with a Hamamatsu Photonics EM-CCD camera, a 100×/1.49NA objective (Olympus) with a

1.6× auxiliary magnification and an environmental chamber (Okolab) to control the temperature. When looking at the effect of TagRFP-HURP on microtubules dynamics (Fig. 4B) and at the binding properties of TagRFP-HURP on barcoded microtubules (Fig. 5), the temperature was kept constant at 30 °C, while when investigating the EGFP-EB3/TagRFP-HURP binding in vitro (Fig. 4A, C) it was set at 35 °C.

## Minimal model of HURP dynamics on K-fibres

In the minimal model of HURP dynamics, we accounted for HURP diffusion, preferential HURP binding to the GDP-lattice and reduced binding of HURP on the GTP-cap, effects of the RanGTP gradient on HURP binding dynamics, and chromosome movements. We took a coordinate system along the K-fibre adjacent to the kinetochore in the direction of the spindle pole (Fig. 6A). Position $x(t)$ thus represents the distance from the kinetochore along a K-fibre of an arbitrary tublin dimer at time $t$. The model takes the form of a continuum partial differential equation with advection, diffusion and reaction terms as follows:

$$\frac{\partial H}{\partial t} + v\frac{\partial H}{\partial x} = D\frac{\partial^2 H}{\partial x^2} + \lambda(x)g(x) - \mu H \tag{2}$$

where $H(x,t)$ is the concentration of HURP, $D$ is the diffusion coefficient of HURP, $\lambda(x)$ is the binding rate, $g(x)$ is the RanGTP gradient, $\mu$ is the unbinding rate, and the speed $v$ is rate of change of the distance between any tubulin dimer and the kinetochore. Due to the kinetochore-centric coordinate system, the advection term $v\frac{\partial H}{\partial t}$ is unaffected by poleward microtubule flux, which occurs due to depolymerisation of microtubule minus ends at the spindle poles but does not affect the distance from the kinetochore to an arbitrary tubulin dimer.

We assume that HURP preferentially binds to GDP-tubulin, and binds at a lower rate to the GTP-cap and the mixed-nucleotide zone. We have incorporated a linear binding profile in space across GTP-cap and mixed-nucleotide zone, meaning that $\lambda(x)$ takes the form:

$$\lambda(x) = \begin{cases} \frac{\lambda}{r} + \left(\lambda - \frac{\lambda}{r}\right)\frac{x}{l} & x \in (0, l) \\ \lambda & x \in (l, L) \end{cases} \tag{3}$$

where $l$ is the length of the GTP-cap and mixed-nucleotide zone, and $r$ is the ratio of binding rates between the regions. Thus the binding rate transitions from $\lambda/r$ at the GTP-cap at $x = 0$ to $\lambda$ at the end of the mixed-nucleotide zone at $x = l$. For leading kinetochores, we can assume $l = 0$ and only the dynamics on the GDP-lattice are relevant. We assume that the RanGTP gradient is stationary and can be described by $g(x) = \exp(-x/s)$, where $s$ is a spatial scale for the RanGTP gradient. Thus, the only difference in the model between leading and trailing kinetochores is the exclusion of HURP in the mixed-nucleotide zone for trailing kinetochores, and the movement of chromosomes in opposite directions relative to the coordinate system. To provide initial conditions, the system is simulated up to equilibrium for the trailing kinetochore, and this equilibrium distribution is used as the initial condition for the leading kinetochore, which is in turn used as the initial condition for simulating the trailing kinetochore.

We assume general Robin boundary conditions (a weighted combination of Dirichlet and Neumann type boundary conditions) at the kinetochore at 0, and at the end of the modelled region of K-fibre at $L$, such that:

$$\begin{aligned} \gamma_1 H(0,t) - D\frac{\partial H(0,t)}{\partial x} &= 0, \\ \gamma_2 H(L,t) - D\frac{\partial H(L,t)}{\partial x} &= 0 \end{aligned} \tag{4}$$

where $\gamma_1, \gamma_2$ are parameters that control the strength of interaction with the boundary. We note that zero flux (Neumann) boundaries are a special case where $\gamma_i = 0$, and fixed (Dirichlet) boundaries similarly

correspond to the case where $\gamma_i \to \infty$. Robin boundary conditions can be interpreted as partially adsorbing boundary conditions such that a particle interacting with the boundary has some probability of reflecting and some probability of adsorbing at the boundary[58].

The model is simulated numerically with parameter values given in Supplementary Table 1 using the MATLAB partial differential equation solver.

### Fitting the minimal HURP model to experimental data via Bayesian inference

A Bayesian inference approach was used to draw samples from the posterior distribution of parameters given observed experimental data via a Markov chain Monte Carlo (MCMC) algorithm. Specifically, we used a Random Walk Metropolis algorithm[59], tuning the covariance of the proposal distribution on a short initial run (1000 iterations).

Let model parameters $\theta = [l, D, \lambda, \mu, \nu_+, \nu_-, \gamma_1, \gamma_2, s, r]$, and observed data $y_+(x_i, t_j)$ and $y_-(x_i, t_j)$ from the trailing and leading K-fibres be available at positions $x_i$ and times $t_j$. We assume that parameters are the same on leading and trailing K-fibres, other than the advection parameter $\nu$ which differs on the trailing (+) and leading (−) K-fibres, and the size of the GTP-cap/mixed-nucleotide zone $l$ which is effectively set to 0 on leading kinetochores. The likelihood for the model forms a product over time and space, using a Gaussian observation model with measurement error, $\sigma$. We assume that the measurement error is known and fixed, and focus on inferring the dynamic model parameters. If $N(z; \mu, \Sigma)$ is the probability density function at location $z$ of a Gaussian distribution with mean $\mu$ and covariance $\Sigma$, then the likelihood is given as:

$$\mathcal{L} = \prod_{k \in \{+, -\}} \prod_{i,j} N(y_k(x_i, t_j); H_k(x_i, t_j), \sigma) \tag{5}$$

where $H_+(x, t)$ or $H_-(x, t)$ are obtained from numerically solving the partial differential equation given by the model for trailing/leading K-fibres, respectively. The grid of positions $x_i$ and times $t_j$ at which the likelihood is evaluated is coarser than the grid over which the differential equation is solved, and corresponds to a point every pixel in space (104 nm) and every frame in time (4.1 s), as dictated by the available data.

The proposal used was a random walk on log space of the parameters as follows for the $i$th parameter:

$$\rho(\theta_i, \Sigma) = sign(\theta_i) \exp(\log(\delta_i + |\theta_i|) + \xi_i) \tag{6}$$

where $\xi \sim N(0,)$ for proposal covariance  tuned on an initial run based on an empirical estimate of the covariance, and $\delta$ is a small tolerance to avoid getting stuck close to 0. We impose priors on these parameters as described in Supplementary Table 1 (weakly informative priors are used for all parameters except for $r$, where a strong prior is used to impose knowledge about relative binding efficiencies based on in vitro data). To initialise the MCMC chains, we draw randomly from the prior.

The MCMC algorithm was run for 10,000 iterations on four chains, with the first half of these discarded as burn-in. On synthetic data (Supplementary Fig. 5A), the MCMC algorithm is able to recover the true parameters used to generate the data (Supplementary Fig. 5B; dashed red lines). Using the observed experimental data, the results are shown via histograms of the marginal posterior distributions for all parameters (Supplementary Fig. 5C). Traceplots (Supplementary Fig. 5D) indicate convergence to the stationary distribution. Software to implement the model and fit this to experimental data is available via https://github.com/shug3502/minimal_HURP_model.

### Total protein extraction

For protein extraction, cells were lysed in NP40 lysis buffer (50 mM Tris-HCl pH 8, 150 mM NaCl, 5 mM EDTA, 1% NP40) supplemented with protease inhibitor (Sigma-Aldrich; 11836170001). Cell lysates were incubated on ice for 15 min and vortexed several times. Cell lysates were centrifuged at 17,000 × g for 10 min at 4 °C and supernatants containing protein extracts were collected.

### Immunoblotting

Protein concentrations were determined using the Bradford Protein Assay (Thermofisher; 23200). Samples of equal amounts of protein were incubated with 5X Laemmli buffer and heated to 95 °C for 5 min. Proteins were separated on a 10% SDS-polyacrylamide gels and transferred onto a nitrocellulose membrane by wet blotting. Membranes were blocked with 5% non-fat dry milk in PBS 0.2% Tween20 (PBS-T) for 30 min. After blocking, membranes were incubated with primary antibodies overnight at 4 °C in PBS-T 5% non-fat dry milk. Membranes were washed three times with PBS-T and incubated 1 h with the appropriate peroxidase-conjugated secondary antibody (1:10,000; goat anti-Mouse; Thermofisher; 32430, goat anti-Rabbit; Thermofisher; G-21234) in PBS-T 5% non-fat dry milk. After three washes, the immunoreactive bands were detected using the Amersham ECL Prime Western Blotting Detection Kit (Cytiva; RPN2232). The following primary antibodies were used: rabbit anti-α-tubulin Ab (1:2000; Abcam; ab18251), rabbit anti-tubulin detyrosinated Ab (1:500; Sigma-Aldrich; AB3201) and mouse anti-actin Ab, clone C4 (1:5000; Sigma-Aldrich; MAB1501R).

### Statistics and reproducibility

Statistical analysis for Figs. 3F, 5B, D and E were performed using GraphPad Prism 8 (GraphPad), the statistical tests employed are described in the figure legends. Analysis of the intensities of HURP and the Ska complex in all in vitro experiments was performed using Excel, while all P values were obtained using Matlab.

Minimum of three independent biological replicates were performed in all experiments. Exception for immunoblotting presented in Fig. 3C (N = 1), representative movies in Fig. 3D (N = 1; 13 and 19 cells for DMSO and PTL treatments, respectively), and Fig. 5F (N = 2), and experiments in Fig. 6E and Supplementary Fig. 4C (N = 1).

### Reporting summary

Further information on research design is available in the Nature Research Reporting Summary linked to this article.

## Data availability

All the raw data related to figures and supplementary figures (representative images and movies) are available on: https://doi.org/10.26037/yareta:ectfpo6p6zfmhoqjpviomcjznq. All other relevant data supporting the key findings of this study are available within the article and its Supplementary Information files or from the corresponding author upon reasonable request. Source data are provided with this paper.

## Code availability

The kinetochore-tracking code is available at https://github.com/cmcb-warwick/KiT and on Zenodo at https://doi.org/10.5281/zenodo.6806435. The FRAP analysis code is available at https://github.com/Bioimaging/FRAP and on Zenodo at https://doi.org/10.5281/zenodo.6810971. The numerical HURP simulation code is available on Github at https://github.com/shug3502/minimal_HURP_model and on Zenodo at https://doi.org/10.5281/zenodo.6617166.

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

## Acknowledgements
We thank A. Khodjakov (New York State Department of Health), P. Nowak-Sliwinska (University of Geneva), A. Straube (Warwick University), E. Nigg (University of Basel), E. Dent (University of Madison) D. Gerlich (IMBA, Vienna), M. Strubin (University of Geneva), D. Trono (EPFL, Lausanne) and T. Surrey (CRG, Barcelona) for reagents. We thank the Bioimaging Facility at the Medical Faculty of the University of Geneva and the Computational & Advanced Microscopy Development Unit at Warwick Medical School for microscopy support; N. Liaudet (University of Geneva) for help in data analysis and writing the FRAP analysis code; I. Gasic (University of Geneva) for discussions and advice about fixation methods; A. Diman and M. Strubin laboratory (University of Geneva) for lentiviruses production support; and members of the Meraldi, McAinsh and Gotta (University of Geneva) laboratories for critical discussions. Work in the PM laboratory is supported by an SNF project grant (No. 31003A_179413) and the University of Geneva. The Lattice Light Sheet Microscope Facility was established at Warwick with a Wellcome Trust Multi-user Equipment grant to A.M. (grant 208384/Z/17/Z). J.H., N.B. and A.M. are supported by BBSRC (BB/R009503/1). A.M. and A.I. are supported by a Wellcome Senior Investigator Award (106151/Z/14/Z).

## Author contributions
The original conceptualisation of the project was by P.M. with subsequent ongoing contribution from A.D.M. C.C. carried out all the cell biology experiments, except Lattice light-sheet microscopy data collection (A.D.M. and O.S.) and relative data analysis. A.V.I. carried out all protein purification, in vitro reconstitution, biochemistry experiments and relative data analysis. Image processing, analysis of lattice data, their simulations and the computational model were carried out by J.U.H. D.D. created EGFP-HURP HeLa cell line and made initial HURP dynamics movies. C.C. and A.V.I. wrote an original draft. A.D.M. and P.M. reviewed and edited the manuscript. A.M., P.M. and N.J.B. acquired funding and supervised the project.

## Competing interests
The authors declare no competing interests.

## Additional information
**Supplementary information** The online version contains

supplementary material available at https://doi.org/10.1038/s41467-022-32421-x.

