## [Peer Review File · Nature Communications]

REVIEWER COMMENTS

Reviewer #1 (Remarks to the Author):

In the manuscript by Castogiovanni, et. al., the authors present evidence that the HURP protein 1) binds to GDP-tubulin in vitro, 2) is a marker for GDP-tubulin within the microtubule lattice, 3) binds to depolymerizing plus ends of MTs, and 4) can act as a rescue factor affecting microtubule dynamics. This is an interesting paper with generally well performed experimentation on an interesting problem. The GTP/GDP makeup of microtubules has been an experimental debate for decades with current models still variable with regards to the number of GTP tubulin dimers make up the growing plus ends (it is reasonably well agreed upon that shrinking plus ends are largely if not completely composed of GDP tubulin). While the title of the paper is focused on the GTP/GDP content of kinetochore MTs, the paper is completely about the HURP protein with some data that it acts as a GDP tubulin binding protein. This puzzling title choice is one of several issues I have with the paper, not the least of which is that I find the conclusions overstated based on the data shown. Below I list additional issue I have with this manuscript:

1) Due to the nature of the problem, nearly all the conclusive work, in my and apparently the field's opinion, has been conducted in vitro supported by chemical analysis such as HPLC to determine nucleotide state. All in vivo work, to my knowledge, has been conducted using EB proteins as a proxy for GTP tubulin. However, EB proteins can and will bind non-GTP microtubules particularly when expressed at high levels. Thus arises one of my main problems with this work, thus use of EB proteins as an indicator of GTP tubulin. This is compounded by the assumption that HURP binding to MTs in vivo is a marker for GDP tubulin. The paper is littered with overstatements derived from the assumptions that these two proteins are absolute markers of either GTP or GDP tubulin, which is a flawed concept. I do not know how one rectifies this other than greatly tempering conclusive statements and pointing out, repeatedly, that interpretations are driven by the X or Y binding constant of EBs or HURP for GTP or GDP tubulin respectfully and that these proteins can and do bind to other forms of tubulin.

2) The images shown are quite hard to interpret. The signals are saturated and the backgrounds appear to have been artificially darkened (by raising the black level). I find it very hard to interpret these data. All the images should be represented without saturation and further manipulation of the background (the JCB has a nice summary on the ethics of image presentation that might be useful). Finally, I was somewhat surprised by the poor quality of the Lattice light sheet data based on all the hype around this instrument, I would recommend the authors turn back to spinning disk as the images there were remarkably better.

3) FRAP was originally presented by Axelrod and Watt Webb in 1976 and has been used ever since as a method to measure protein dissociation constants in living systems. FRAP does not measure "on-rates" of proteins (association constants) as the authors posit. This should be fixed or removed from the paper.

4) A recent paper by the Roll-Mecak lab (Science August 2021) presents really beautiful work on katanin severing MTs through the lattice and incorporating new, GTP tubulin. I realize that this paper was most likely prepared prior to publication of this work, however feel like it should be addressed in a revised version.

5) Finally, the title of this paper is misleading (a pointed out above) and should be changed to reflect the work presented on the HURP protein.

Reviewer #2 (Remarks to the Author):

The mitotic spindle is a complex structure that orchestrate forces in space and time to accurately segregate genetic material. This neatly organized machine relies on the coordinated action of numerous proteins that work together to accomplish this important biological process. For example, hepatoma upregulated protein (HURP) accumulates near chromosomes to form bundled kinetochore microtubules (K-fibers). In this joint study, the authors combine in vivo and in vitro experiments with theoretical modeling. They investigate distributions of HURP in space and time and find that this protein accumulates along depolymerizing K-fibers but does not accumulate at the tip region of the polymerizing K-fibers. Based on this observation, together with additional in vivo and in vitro experiments and mathematical modeling, the authors conclude that K-fibers contain a micron-sized mixed-nucleotide zone. However, this study also contains some weaknesses that need to be addressed.

Major criticism:

1. To test their hypothesis, the authors should provide a more coherent investigation of the distributions of HURP in space and time, where experiment and theory would be more interwoven. This would imply a closer comparison of quantitative measurements and theoretical calculations. Here are several suggestions for improvement of the manuscript (see also points 2. – 5.): The authors performed FRAP experiment and found that recovery half-time is around 10 s. Based on this result, the authors directly estimate an on-rate of HURP. However, HURP recovery is not driven by attachment of cytoplasmic molecules only. For example, diffusion of HURP along the microtubule could also contribute to HURP signal recovery. Because the mathematical model describes HURP dynamics in more details, the authors should simulate FRAP experiment with the mathematical model and extract the unknown model parameter(s) by comparison with FRAP experiment.

2. In the mathematical model, size of GTP-cap and mixed-nucleotide zone is fixed to 0.5 microns. However, this parameter is directly related to the hypothesis of this study and thus deserves more attention. Therefore, it would be important to explore the model by varying this parameter. In particular, it would be interesting to test what happens in the absence of the mixed-nucleotide zone, i.e.

when this parameter is 0 microns. Based on the outcome, the authors could propose additional predictions or, even, rule out some alternative explanations.

3. In the mathematical model, there is a drift term which describes the directed movement of HURP with respect to kinetochores. Depending on whether the kinetochores move towards or away from the pole (leading/trailing kinetochores), the parameter representing the velocity of kinetochores has a negative or positive sign. However, in the mitotic spindle, in addition to movement of kinetochores, K-fibers undergo poleward flux. Because poleward flux could influence the distributions of HURP, and consequently the final conclusions, poleward flux should be included in the mathematical model. This can be simply incorporated by using a different choice of parameters that represent the velocity of the trailing and leading kinetochores by adding a constant value (poleward flux velocity) to these two parameters.

4. The model has 9 parameters, but the authors provide a reference for only 2 parameters. Because the conclusions depend on the parameter values, the authors should better justify their choice. For example, if the authors measured some of these parameters in this study, they could put that information in Table S1. On the other hand, if some of these parameters are free, the authors should use it as an adjustable parameter and explore the model more thoroughly by varying them (and state that in Table S1 as well).

5. In the model, diffusion of HURP along the microtubule could influence redistribution of HURP and consequently change its distributions. Thus, diffusion coefficient is an important parameter, and changes in its value could affect the main predictions of the model. To narrow the values of the diffusion coefficient parameters, the authors could revisit their in vitro experiments to measure or estimate HURP diffusion coefficient in vitro.

6. The authors should also discuss alternative mechanisms. One alternative mechanism could be related to the difference between tyrosinated and detyrosinated microtubules (Barisic et al., Science 2015). E.g. the observed HURP free zone could arise from different affinity of HURP to tyrosinated and detyrosinated microtubules.

Reviewer #3 (Remarks to the Author):

In this manuscript, the authors study the appearance of HURP gaps on kinetochore microtubules (KMTs) in RPE1 cells using quantitative live cell microscopy. The authors observe that HURP, a protein they find preferentially binds to GDP-tubulin and accumulates on long-lived KMTs but does not bind a region near

the end of polymerizing KMTs giving rise to an HURP gap. As KMTs depolymerize, the HURP gap shrinks. The authors find the HURP increases microtubule rescue frequencies in-vitro and that both EB3 and HURP are excluded from a a micron sized zone between the EB3 positive cap at the plus ends of kinetochore microtubules and the accumulated HURP stripes. They interpret this gap as a mixed-nucleotide zone and argue via simulations that this minimal model can reproduce the observed behavior. Overall, the paper is well written and the experiments are a significant step towards a better understanding of kinetochore microtubule dynamics and metaphase oscillations in human cells; however, the role of the proposed mixed nucleotide zone is not entirely clear and should be clarified in the text as well as explored in more details in the computational modeling prior to publication.

Major Concerns:

1. There is no direct evidence offered for the existence of a mixed nucleotide zone on KMTs as appears in Figure 5C. The authors show that HURP preferentially binds to GDP tubulin and EB3 preferentially binds to GTP tubulin, and infer that the gap region between them must be the result of a zone with mixed GTP and GDP tubulin. This could instead result from post-translational tubulin modifications or other regulatory mechanisms. The mixed nucleotide zone would be better phrased as a HURP/EB3 excluded zone.
2. In the modeling associated with Figure 5, the authors take the binding rate in the HURP exclusion zone $\alpha=0$; however, in the data shown in Figure 2C, 3C and 3E indicate that the transition from the gap to the HURP stripe is gradual. The mutant tubulin experiment in Figure 4F indicates that the reduced HURP tubulin affinity, while statistically significant, is quite modest (~70%). Similarly, the reduction in HURP intensity between the gap and the stripe appears to be on the order of 20-30% as opposed to a sharp drop to zero HURP. It would be interesting to see how the simulated curves change treating the exclusion zone with differing HURP tubulin affinity profiles $\alpha(x)$, for example a linear increase over the course of the zone or simply a uniform non-zero α .
3. The authors use fluorescence intensity as a proxy for concentration of the tagged protein in comparing the affinity of HURP for different forms of GTP and GDP tubulin (Figure 2H, 4F) ; however, in over-expression systems there is significant cell to cell variation of the tagged molecules making this intensity comparison difficult to interpret. Correcting for varying expression by normalizing by the cytoplasmic background intensity could correct for the intensity variation.

Minor Concerns:

1. The details in the modeling section of the methods could be expanded for greater clarity
2. It is not entirely clear what the contribution of the background signal is to the HURP intensities plotted throughout the paper. For example, in the modeling/data comparisons in Figure 5 the HURP intensity drops to zero in the model predictions but shows a more modest decline of 20-30% from 0.65 to 0.5. It is unclear whether this is the result of fluorescence background in the data or a more glaring discrepancy between the data and the model.

Point-by-point rebuttal letter

We were pleased that the reviewers generally liked our work and have addressed their specific concerns in the following manner:

Reviewer #1 (Remarks to the Author):

In the manuscript by Castogiovanni, et. al., the authors present evidence that the HURP protein 1) binds to GDP-tubulin in vitro, 2) is a marker for GDP-tubulin within the microtubule lattice, 3) binds to depolymerizing plus ends of MTs, and 4) can act as a rescue factor affecting microtubule dynamics. This is an interesting paper with generally well performed experimentation on an interesting problem. The GTP/GDP makeup of microtubules has been an experimental debate for decades with current models still variable with regards to the number of GTP tubulin dimers make up the growing plus ends (it is reasonably well agreed upon that shrinking plus ends are largely if not completely composed of GDP tubulin). While the title of the paper is focused on the GTP/GDP content of kinetochore MTs, the paper is completely about the HURP protein with some data that it acts as a GDP tubulin binding protein. This puzzling title choice is one of several issues I have with the paper, not the least of which is that I find the conclusions overstated based on the data shown. Below I list additional issue I have with this manuscript:

We appreciate the reviewers concern. We have adapted the title to: Evidence for a HURP/EB free mixed-nucleotide zone in kinetochore-microtubules. We feel that this title still reflects the broader interest of our work which reveals that k-fibres contain a third, unexpected region between the GTP-cap and the GDP-tubulin lattice whilst being more specific.

1) Due to the nature of the problem, nearly all the conclusive work, in my and apparently the field's opinion, has been conducted in vitro supported by chemical analysis such as HPLC to determine nucleotide state. All in vivo work, to my knowledge, has been conducted using EB proteins as a proxy for GTP tubulin. However, EB proteins can and will bind non-GTP microtubules particularly when expressed at high levels. Thus arises one of my main problems with this work, thus use of EB proteins as an indicator of GTP tubulin. This is compounded by the assumption that HURP binding to MTs in vivo is a marker for GDP tubulin. The paper is littered with overstatements derived from the assumptions that these two proteins are absolute markers of either GTP or GDP tubulin, which is a flawed concept. I do not know how one rectifies this other than greatly tempering conclusive statements and pointing out, repeatedly, that interpretations are driven by the X or Y binding constant of EBs or HURP for GTP or GDP tubulin respectfully and that these proteins can and do bind to other forms of tubulin.

While agree with the reviewer that EBs and HURP cannot be considered as absolute markers for GTP and GDP microtubules respectively, we feel that our main conclusion (the presence of a mixed-nucleotide zone) is supported by a strong set of data.

- 1) Although EB is a widely recognized marker for GTP-tubulin, we agree that overexpressed EB3 could sometimes also recognize non-GTP tubulin microtubules (there can be false positives). This, however, also means that the absence of overexpressed EB3 in the HURP-gaps can only indicate that such a zone is not composed of pure GTP-tubulin.*
- 2) Similarly, our in vitro and in vivo data strongly indicate that HURP (which in this case present at endogenous levels) very efficiently binds to the GDP-tubulin lattice. Therefore, lack of HURP only leaves the conclusion that the HURP- gap is not composed of pure GDP-tubulin.*
- 3) Given these two constraints, the only remaining conclusion is that the HURP-gap reflects a mixed-nucleotide zone.*

2) The images shown are quite hard to interpret. The signals are saturated and the backgrounds appear to have been artificially darkened (by raising the black level). I find it very hard to interpret these data. All the images should be represented without saturation and further manipulation of the background (the JCB has a nice summary on the ethics of image presentation that might be useful). Finally, I was somewhat surprised by the poor quality of the Lattice light sheet data based on all the hype around this instrument, I would recommend the authors turn back to spinning disk as the images there were remarkably better.

We thank the reviewer for pointing out the high background and/or signal saturation in some of our figure panels. We have reviewed and adjusted the contrast for a better visual representation, most notably in Fig. 1E, 1I; Supplementary Fig. 1C and Fig. 3E. Figure 1G image was acquired with a spinning disk microscope and was previously presented in a raw state. In order to improve the rendering, we have now deconvolved all channels in the new version of the figure 1G. We hope that these modifications will now give full satisfaction. Nevertheless, we also emphasize that we were already following the JCB guidelines, as we had indicated modification of the gamma-value. We also wish to highlight that lattice light sheet microscopy (LLSM) imaging strengths lie specifically on the greatly improves axial resolution (~308 nm in z in our experiments), negligible out-of-focus background and dramatic reduction in photo-bleaching/toxicity damage (Chen et al., Science, 2014). Simultaneous illumination of the entire field of view in each image plane permits 3D imaging at hundreds of planes per second, and allows visualization of intracellular processes at unprecedented spatiotemporal resolution, providing an immediate advantage for 3D live-cell imaging over any current technology, including spinning disk (affected by a lower axial resolution and slower imaging, see Wang et al., J. Microscopy, 2005). We have now expanded the original figure to detail the 3D acquisition parameters and full z-stack visualization, highlighting the benefits of LLSM (please see Fig. 2A); we also improved the panels view of the representative cell to only show 11 z-slices, increasing the overall clarity (please see Fig. 2B). We believe the new figure provides a better visual representation of the lattice data.

3) FRAP was originally presented by Axelrod and Watt Webb in 1976 and has been used ever since as a method to measure protein dissociation constants in living systems. FRAP does not measure “on-rates” of proteins (association constants) as the authors posit. This should be fixed or removed from the paper.

We thank the reviewer for pointing out this imprecise formulation. We have removed the term 'on-rate' from the text and we now state that our FRAP experiments allows us to monitor the rate at which HURP is recruited from the cytoplasm or by diffusion from neighboring HURP-stripes (taking in consideration that there is no immobile HURP fraction). We hope that will make it easier for readers to understand.

4) A recent paper by the Roll-Mecak lab (Science August 2021) presents really beautiful work on katanin severing MTs through the lattice and incorporating new, GTP tubulin. I realize that this paper was most likely prepared prior to publication of this work, however feel like it should be addressed in a revised version.

We thank the reviewer for pointing out this study. We have now specified in the introduction that GTP-tubulin is not only found at the MT plus-end tips, but also along microtubule shafts as a part of microtubule self-repair. We have also included additional references on this matter.

5) Finally, the title of this paper is misleading (a pointed out above) and should be changed to reflect

See our comment on point 1

Reviewer #2 (Remarks to the Author):

The mitotic spindle is a complex structure that orchestrate forces in space and time to accurately segregate genetic material. This neatly organized machine relies on the coordinated action of numerous proteins that work together to accomplish this important biological process. For example, hepatoma upregulated protein (HURP) accumulates near chromosomes to form bundled kinetochore microtubules (K-fibers). In this joint study, the authors combine in vivo and in vitro experiments with theoretical modeling. They investigate distributions of HURP in space and time and find that this protein accumulates along depolymerizing K-fibers but does not accumulate at the tip region of the polymerizing K-fibers. Based on this observation, together with additional in vivo and in vitro experiments and mathematical modeling, the authors conclude that K-fibers contain a micron-sized mixed-nucleotide zone. However, this study also contains some weaknesses that need to be addressed.

Major criticism:

1. To test their hypothesis, the authors should provide a more coherent investigation of the distributions of HURP in space and time, where experiment and theory would be more interwoven. This would imply a closer comparison of quantitative measurements and theoretical calculations. Here are several suggestions for improvement of the manuscript (see also points 2. – 5.): The authors performed FRAP experiment and found that recovery half-time is around 10 s. Based on this result, the authors directly estimate an on-rate of HURP. However, HURP recovery is not driven by attachment of cytoplasmic molecules only. For example, diffusion of HURP along the microtubule could also contribute to HURP signal recovery. Because the mathematical model describes HURP dynamics in more details, the authors should simulate FRAP experiment with the mathematical model and extract the unknown model parameter(s) by comparison with FRAP experiment.

To address this point, we have made substantial modifications to the computational modelling part of our work, by fully parameterising the model in a Bayesian framework using experimental data derived from the in vivo measurements in Fig. 2D. This approach allows us to estimate directly all the model parameters in the cellular environment, including the diffusion coefficient. A full description is provided in the methods and results are shown in Fig. 6, including the marginal distributions estimated for key model parameters. To further validate the computational model and this fitting procedure, we have simulated a FRAP experiment using the parameterised model (Fig. 6F) and obtain a timescale for recovery of HURP signal similar to that observed in vivo. Moreover, we clarified in the text that in the FRAP experiment HURP "repopulates a FRAP-zone on single K-Fibres from the cytoplasm or through HURP diffusion along microtubules."

2. In the mathematical model, size of GTP-cap and mixed-nucleotide zone is fixed to 0.5 microns. However, this parameter is directly related to the hypothesis of this study and thus deserves more attention. Therefore, it would be important to explore the model by varying this parameter. In particular, it would be interesting to test what happens in the absence of the mixed-nucleotide zone, i.e. when this parameter is 0 microns. Based on the outcome, the authors could propose additional predictions or, even, rule out some alternative explanations.

We thank the reviewer for highlighting the importance of this parameter in the mathematical model. To investigate further the sensitivity of the model to the size of the MNZ, we have varied this parameter and shown the effect on the HURP gap (Fig. 6D). Additionally, we have added plots to the supplementary material to show the effect on the full spatiotemporal dynamics of varying the size of the MNZ (including with this size set to 0) with other parameters fixed. We further note that, although this comparison is performed with all other parameters fixed, when fitting the model to experimental data, all parameters are allowed to vary jointly such that this fitting procedure considers the case with of a MNZ of size 0, but finds it unlikely compared to the larger values obtained in the posterior distribution (Fig. 6B).

3. In the mathematical model, there is a drift term which describes the directed movement of HURP with respect to kinetochores. Depending on whether the kinetochores move towards or away from the pole (leading/trailing kinetochores), the parameter representing the velocity of kinetochores has a negative or positive sign. However, in the mitotic spindle, in addition to movement of kinetochores, K-fibers undergo poleward flux. Because poleward flux could influence the distributions of HURP, and

consequently the final conclusions, poleward flux should be included in the mathematical model. This can be simply incorporated by using a different choice of parameters that represent the velocity of the trailing and leading kinetochores by adding a constant value (poleward flux velocity) to these two parameters.

The drift term in the model does describe directed movement of HURP with respect to kinetochores. More specifically, the drift term gives the rate of change of the distance between any tubulin dimer and the kinetochore. Poleward flux occurs due to depolymerisation of microtubule minus ends at the spindle poles. However, this does not then affect the distance between a tubulin dimer and the kinetochore (and associated HURP dynamics). Thus our model is independent of poleward flux. We have added additional text to the methods section to highlight this point. Nevertheless, on a theoretical basis one could assume that the length of the HURP-gap correlates with flux rates, as cells need to increase plus-end polymerization to compensate for minus-end depolymerization at steady-state spindle length. We therefore decided to experimentally manipulate the rate of minus-end depolymerization in cells and monitor the size of the HURP-gaps. As the most specific tool we were aware of, we used the depletion of WDR62, a spindle pole protein which we recently found to be required for efficient poleward microtubule flux due to its role as recruitment platform for Katanin at spindle poles (Guerreiro A. et al., JCB (2021); but see also Huang et al., JCB in the same issue). Indeed, WDR62 depletion reduces flux rates by half in our hands. We thus first validated by immunofluorescence and by immunoblotting (Figure 1 for reviewers A and B) the efficiency of WDR62 depletion in hTERT-RPE1 EGFP-HURP/Halo-CENP-A cells. Next we monitored the size of the HURP-gaps after WDR62 depletion, and observed a slight increase in their size (Figure 1 for reviewers C and D and MOVIE_WDR62_for_reviewers). WDR62 depletion, however, also slightly increases the length of the mitotic spindle length, the speed of kinetochores and the amplitude the sister-kinetochore oscillations (Figure 1 for reviewers E-H and Guerreiro A. et al., JCB (2021)). Since the latter two parameters would be expected to increase the size of HURP-gaps, we feel that these results are inconclusive: whichever way one wants to interpret these results one can find an argument in favor or against it. The only conclusion that can be drawn is that there is no strong correlation between the HURP-gap and flux rates. Given these major uncertainties we decided not to include these results, but to only present it to the reviewers to share this information.

Figure 1 for reviewers

Guerreiro A. et al., JCB (2021)

Reduction of poleward microtubule flux after WDR62 depletion slightly increase the size of the MNZ. (A) Representative immunofluorescence images of metaphase hTERT-RPE1 EGFP-HURP/HaloTag-CENP-A cells transfected for 48 h with siCTRL or siWDR62 and stained for WDR62, α -tubulin and DAPI. **(B)** Western blot analysis of WDR62 levels in hTERT-RPE1-EGFP-HURP/HaloTag-CENP-A cells transfected for 48 h with siCTRL or siWDR62. **(C)** Live cell imaging of metaphase hTERT-RPE1 EGFP-HURP/HaloTag-CENP-A cells transfected for 48 h with siCTRL or siWDR62. Inset shows sister K-fibre pair used for kymograph (right panel), bars display the HURP-gaps maximum length. Scale bars = 5 μ m and 1 μ m (kymograph). **(D)** Distribution of live HURP-gap maximum lengths in hTERT-RPE1 EGFP-HURP/HaloTag-CENP-A cells transfected for 48 h with siCTRL or siWDR62. ($p = t$ -test; $N = 2$ independent experiments, siCTRL; $n = 84$ K-fibers in 30 cells, siWDR62; $n = 91$ K-fibers in 26 cells). Green and blue lines = curve fit. **(E)** Quantification of pole-to-pole distances in siCTRL-, siKATNB1-, and siWDR62-treated RPE1 metaphase cells. Dot plots show medians per experiment; curves the frequency distribution; bars represent mean \pm SEM; $**$, $P < 0.01$ in two-tailed unpaired t test. **(F)** Schematic illustrating the parameters of sister-kinetochore oscillation autocorrelation curves. **(G)** Autocorrelation curves of sister-kinetochore oscillations along spindle axis. **(H)** Sister-kinetochore velocities. $N = 3-5$, $n = 35$ (siCTRL), 12 (siCAPD2), 34 (siKATNB1), and 33 (siWDR62) cells; dot plots represent each cell with bars displaying mean \pm SEM; box plot represent mean \pm SD; $*$, $P < 0.05$; $****$, $P < 0.0001$, one-way ANOVA. AU, arbitrary units. **E-F**, Data and figures are from Guerreiro A. et al., JCB (2021).

4. The model has 9 parameters, but the authors provide a reference for only 2 parameters. Because the conclusions depend on the parameter values, the authors should better justify their choice. For example, if the authors measured some of these parameters in this study, they could put that information in Table S1. On the other hand, if some of these parameters are free, the authors should use it as an adjustable parameter and explore the model more thoroughly by varying them (and state that in Table S1 as well).

We have addressed this point by performing full parameterisation of the model in a Bayesian framework, as outlined previously in the response to point 1. This ensures that estimates of the model parameters are calibrated to experimental measurements from in vivo data. We have updated Table S1 to include our estimates of the model parameters, and the uncertainty in these estimates.

5. In the model, diffusion of HURP along the microtubule could influence redistribution of HURP and consequently change its distributions. Thus, diffusion coefficient is an important parameter, and changes in its value could affect the main predictions of the model. To narrow the values of the diffusion coefficient parameters, the authors could revisit their in vitro experiments to measure or estimate HURP diffusion coefficient in vitro.

We thank the reviewer for this suggestion. We have obtained experimental estimates of the diffusion coefficient in vitro. To ensure sufficient temporal resolution to analyse the diffusion of HURP on microtubules, we collected additional movies at higher temporal resolution. These movies of dynamic microtubules in the presence of 0.3 nM TagRFP-HURP at 35°C enable capture of single molecule events. This new dataset has been added in the main manuscript (Fig. 6E and S4B) and the estimated diffusion coefficient agrees with the value found from fitting our mathematical model to the lattice light sheet data (Fig. 6B and Table S1). Furthermore, we used this new high temporal resolution data to revisit the analysis of the HURP lifetimes on GDP and GMPCPP microtubule lattice and, despite not finding any differences with the previous dataset, we have now replaced the graphs in Fig. S4C, to reflect the newest data.

6. The authors should also discuss alternative mechanisms. One alternative mechanism could be related to the difference between tyrosinated and detyrosinated microtubules (Barisic et al., Science 2015). E.g. the observed HURP free zone could arise from different affinity of HURP to tyrosinated and detyrosinated microtubules.

We thank the reviewer for this suggestion. Indeed, in the previous version of the manuscript, the impact of tyrosinated/detyrosinated tubulin on HURP binding was not investigated, since we could reproduce all the in vivo observations in our minimal in vitro microtubule assays. Those in vitro assays were performed using either purified pig brain tubulin (Fig.4 in the new version of the manuscript), corresponding to a mixture of tubulin with different post-translational modification or with in vitro purified human tubulin which only contains the tyrosinated form of α -tubulin (Fig.5 in the new version of the manuscript). In those conditions we saw no differences in the readout, suggesting that HURP binding is independent on tubulin modifications. Nevertheless, to further test the potential role of tubulin detyrosination, we treated cells with Parthenolide, an inhibitor of tubulin detyrosination. To facilitate live cell-imaging, a co-treatment with MG132 was used to enrich cells in metaphase as described in Barisic M., et al., Science (2015). We observed that the increase of tyrosinated tubulin in cells did not increase the size of HURP-gaps, as one could have hypothesized in case HURP would have a low affinity for freshly incorporated tyrosinated tubulin on growing K-fibres. These new data are included in Figure 3.

Reviewer #3 (Remarks to the Author):

In this manuscript, the authors study the appearance of HURP gaps on kinetochore microtubules (KMTs) in RPE1 cells using quantitative live cell microscopy. The authors observe that HURP, a protein they find preferentially binds to GDP-tubulin and accumulates on long-lived KMTs but does not bind a region near the end of polymerizing KMTs giving rise to an HURP gap. As KMTs depolymerize, the HURP gap shrinks. The authors find the HURP increases microtubule rescue frequencies in-vitro and that both EB3 and HURP are excluded from a micron sized zone between the EB3 positive cap at the plus ends of kinetochore microtubules and the accumulated HURP stripes. They interpret this gap as a mixed-nucleotide zone and argue via simulations that this minimal model can reproduce the observed behavior. Overall, the paper is well written and the experiments are a significant step towards a better understanding of kinetochore microtubule dynamics and metaphase oscillations in human cells; however, the role of the proposed mixed nucleotide zone is not entirely clear and should be clarified in the text as well as explored in more details in the computational modeling prior to publication.

Major Concerns:

1. There is no direct evidence offered for the existence of a mixed nucleotide zone on KMTs as appears in Figure 5C. The authors show that HURP preferentially binds to GDP tubulin and EB3 preferentially binds to GTP tubulin, and infer that the gap region between them must be the result of a zone with mixed GTP and GDP tubulin. This could instead result from post-translational tubulin modifications or other regulatory mechanisms. The mixed nucleotide zone would be better phrased as a HURP/EB3 excluded zone.

We appreciate the reviewer concern, but feel that our data allow us to exclude these possibilities. First, as pointed out in Point 1 of reviewer 1 the fact that the HURP-gap contains neither EB3 nor HURP, strongly suggests that this zone cannot contain only pure GTP- or GDP-tubulin microtubules. Second, our new data testing the contribution of tubulin dephosphorylation/tyrosination indicates that this post-translational modification which is differentially present on growing k-fibres cannot explain the HURP-gaps. Third, the only other post-translational modification that is known to be differentially present on stable K-fibres versus freshly polymerized microtubules is acetylated tubulin. In this case, however, we would expect that high taxol doses, which are known to increase tubulin acetylation (Szyk A. et al., Cell (2014); Piperno G. et al., J. Cell Biol (1987); Xiao H. et al., PNAS (2006)), should lead to higher HURP levels. However, as we now explain in our discussion, we observed the opposite effect, a reduction in HURP levels, which excludes tubulin acetylation as a cause for the HURP-pattern.

2. In the modeling associated with Figure 5, the authors take the binding rate in the HURP exclusion zone $I=0$; however, in the data shown in Figure 2C, 3C and 3E indicate that the transition from the gap to the HURP stripe is gradual. The mutant tubulin experiment in Figure 4F indicates that the reduced HURP tubulin affinity, while statistically significant, is quite modest (~70%). Similarly, the reduction in HURP intensity between the gap and the stripe appears to be on the order of 20-30% as opposed to a sharp drop to zero HURP. It would be interesting to see how the simulated curves change treating the exclusion zone with differing HURP tubulin affinity profiles $I(x)$, for example a linear increase over the course of the zone or simply a uniform non-zero I .

To better capture the reduced HURP tubulin affinity in the modelling, we made alterations to the model such that the binding rate in the mixed nucleotide zone follows a linear binding rate in space (see Methods for further details). Parameter values for the binding rate in this region (and all other parameters) are estimated from experimental data in a Bayesian framework and used to simulate from the model. In addition, we demonstrate the sensitivity to changing such assumptions about the difference in binding between these regions in supplementary material, including treating the HURP binding rate as zero, or as a non-zero constant over the mixed nucleotide zone (Figure 2 for Reviewers). The linear binding profile is used in the main text as this results in a smooth distribution of HURP in space in simulations, consistent with observed data. Discontinuities in the HURP binding affinity result in a non-smooth distribution of HURP in space. It is also worth noting though that the quantification of the mutant tubulin experiment in the old Fig. 4F (Fig. 5F in the revised manuscript) is an underestimation of the E254A mutant effect on HURP binding, as it is from a composite of cells expressing the mutant at different levels, as can be seen in the panel with the representative cells and in Fig. S4E. Further-

more, ensuring that the data in Fig. 2 are background subtracted highlights the reduction in HURP intensity between the gap and the stripe.

Figure 2 for reviewers

A

Zero constant binding

B

Non-zero constant binding

C

Linear binding

Effect of binding affinity assumptions in the mixed nucleotide zone (MNZ) and GTP-cap region. (A) Simulations obtained from a model with zero binding in the MNZ showing the dynamics of HURP in space and time on trailing and leading K-fibres. Parameters are obtained by fitting this zero binding model to experimental data. **(B)** Simulations obtained from a model with non-zero constant binding in the MNZ showing the dynamics of HURP in space and time on trailing and leading K-fibres. Parameters are obtained by fitting this non-zero binding model to experimental data. **(C)** Simulations obtained from a model with HURP binding affinity assumed to be linear in space across the MNZ, as described in Methods. Dynamics of HURP are shown in space and time on trailing and leading K-fibres. Parameters are obtained by fitting this linear binding model to experimental data.

3. The authors use fluorescence intensity as a proxy for concentration of the tagged protein in comparing the affinity of HURP for different forms of GTP and GDP tubulin (Figure 2H, 4F); however, in over-expression systems there is significant cell to cell variation of the tagged molecules making this intensity comparison difficult to interpret. Correcting for varying expression by normalizing by the cytoplasmic background intensity could correct for the intensity variation.

We thank the reviewer for this comment and have now ensured that background subtraction has been applied in all analysis of fluorescence intensity and plotting of intensity. The analysis in the mentioned figures are based on the presence of endogenously tagged EGFP-HURP. This cell line has been previously generated by CRISPR knock-in (see details in Dudka et al. Current Biol (2019)), therefore it is not an over-expression system.

Minor Concerns:

1. The details in the modeling section of the methods could be expanded for greater clarity

We have made extensive changes to the modelling section of the methods and the main text to give greater detail and clarity on the mathematical model, and have added a further section to the methods on fitting the model to experimental data.

2. It is not entirely clear what the contribution of the background signal is to the HURP intensities plotted throughout the paper. For example, in the modeling/data comparisons in Figure 5 the HURP intensity drops to zero in the model predictions but shows a more modest decline of 20-30% from 0.65 to 0.5. It is unclear whether this is the result of fluorescence background in the data or a more glaring discrepancy between the data and the model.

In the previous version of this work, the background signal had not been subtracted from the HURP intensities. As discussed in the reviewer's major concern point 3, we have now addressed this by ensuring that the HURP background signal is estimated in each cell and subtracted for all kinetochore trajectories analysed from that cell. The revised version of the same figure (Fig. 6C) shows a much closer match in this aspect between the data with background subtracted, and the simulations from the calibrated model.

REVIEWERS' COMMENTS

Reviewer #1 (Remarks to the Author):

The Authors have adequately addressed all of my concerns and I suggest acceptance for publication.

Reviewer #2 (Remarks to the Author):

The authors addressed all my concerns adequately and I have no further questions.

Reviewer #3 (Remarks to the Author):

The author's have adressed my concerns